# Thresholds for post-rebound SHIV control after CCR5 gene-edited autologous hematopoietic cell transplantation

E Fabian Cardozo-Ojeda[1]*, Elizabeth R Duke[1,2], Christopher W Peterson[2,3,4], Daniel B Reeves[1], Bryan T Mayer[1], Hans-Peter Kiem[2,3,4,5], Joshua T Schiffer[1,2,3]*

[1]Vaccine and Infectious Disease Division, University of Washington, Seattle, United States; [2]Department of Medicine, University of Washington, Seattle, United States; [3]Clinical Research Division, Fred Hutchinson Cancer Research Center, Seattle, United States; [4]Stem Cell and Gene Therapy Program, Fred Hutchinson Cancer Research Center, Seattle, United States; [5]Department of Pathology, University of Washington, Seattle, United States

**Abstract** Autologous, CCR5 gene-edited hematopoietic stem and progenitor cell (HSPC) transplantation is a promising strategy for achieving HIV remission. However, only a fraction of HSPCs can be edited ex vivo to provide protection against infection. To project the thresholds of CCR5-edition necessary for HIV remission, we developed a mathematical model that recapitulates blood T cell reconstitution and plasma simian-HIV (SHIV) dynamics from SHIV-1157ipd3N4-infected pig-tailed macaques that underwent autologous transplantation with CCR5 gene editing. The model predicts that viral control can be obtained following analytical treatment interruption (ATI) when: (1) transplanted HSPCs are at least fivefold higher than residual endogenous HSPCs after total body irradiation and (2) the fraction of protected HSPCs in the transplant achieves a threshold (76–94%) sufficient to overcome transplantation-dependent loss of SHIV immunity. Under these conditions, if ATI is withheld until transplanted gene-modified cells engraft and reconstitute to a steady state, spontaneous viral control is projected to occur.

*For correspondence:
ecojeda@fredhutch.org (EFC-O);
jschiffe@fredhutch.org (JTS)

## Introduction

The major obstacle to HIV-1 eradication is a latent reservoir of long-lived, infected cells (*Chun et al., 1997*; *Chun et al., 1995*; *Finzi et al., 1997*). Cure strategies aim to eliminate all infected cells or permanently prevent viral reactivation from latency. The only two known cases of HIV cure, the 'Berlin Patient' and 'London Patient', resulted from allogeneic hematopoietic stem cell (HSC) transplant with homozygous CCR5Δ32 donor cells (*Allers et al., 2011*; *Hütter et al., 2009*; *Gupta et al., 2019*; *Gupta et al., 2020*), a mutation that makes cells resistant to CCR5-tropic HIV-1. The Berlin Patient was diagnosed with HIV in 1995 and received total body irradiation and allo-HSC transplantation for the treatment of his acute myeloid leukemia in 2007 and 2008. On the day of his first transplantation, antiretroviral therapy (ART) was interrupted, and HIV viremia never returned (*Allers et al., 2011*; *Hütter et al., 2009*; *Peterson and Kiem, 2019*). In 2019, an HIV-1 remission for more than 18 months was reported in the London Patient as part of the IciSTEM cohort (*Gupta et al., 2019*). The London Patient underwent one allo-HSC transplantation for treatment of Hodgkin Lymphoma in 2016, but with a less aggressive conditioning compared to the Berlin patient without irradiation (*Gupta et al., 2019*). This individual stopped ART 17 months after transplantation and as of March, 2020 his viremia remains suppressed, representing a possible case of HIV-1 cure (*Gupta et al., 2020*). The success of the allo-HSC transplantation is likely multifactorial—in part attributable to HIV resistance of the transplanted cells, the conditioning regimen that facilitates engraftment and

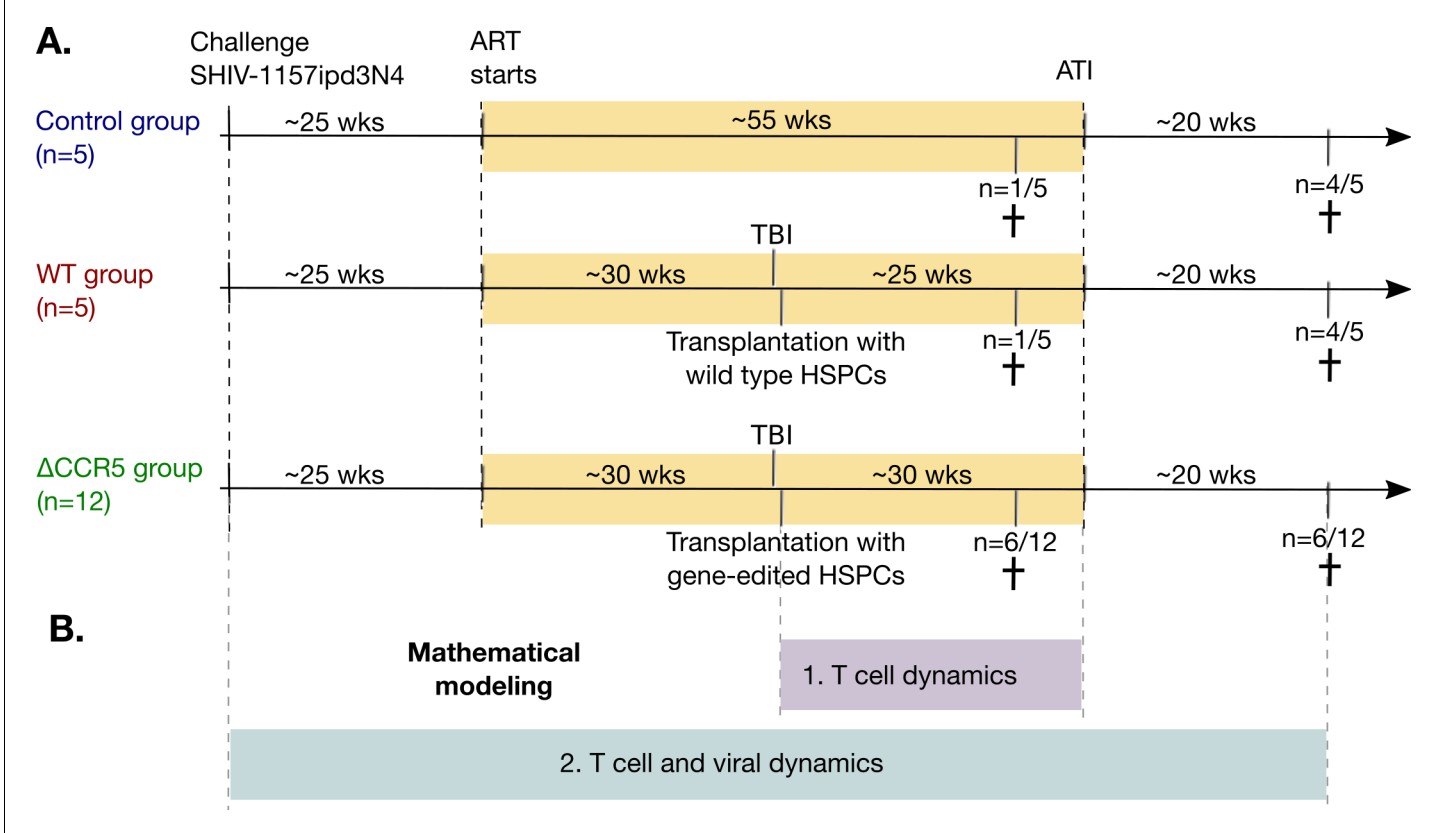

**Figure 1.** Study design and mathematical modeling. (**A**) Twenty-two pig-tailed macaques were infected with SHIV and suppressed with ART. Next, 17/22 underwent hematopoietic stem and progenitor cell (HSPC) transplantation following myeloablative conditioning (TBI), including 12 animals that received CCR5-edited products and five that received non-edited products (ΔCCR5 and WT groups, respectively). A control group (n = 5) did not receive TBI or HSPC transplantation. Fourteen animals underwent ATI approximately 1 year after ART initiation, while the remaining eight animals were necropsied prior to ATI (see Materials and methods for details). (**B**) We first developed mathematical models for T cell dynamics and reconstitution following transplant and before ATI (purple), assuming that low viral loads on ART do not affect cell dynamics. After validation of that model, we introduced viral dynamics and fit those to the T cell, primary infection, and viral rebound dynamics from the animals pre- and post-ATI (blue).

eliminates infected cells, graft-versus-host effect against residual infected cells, and immunosuppressive therapies for graft-versus-host disease (*Henrich et al., 2016*; *Henrich et al., 2014*; *Henrich et al., 2013*; *Salgado et al., 2018*).

We are interested in recapitulating this method of cure but with reduced toxicity. Specifically, we are investigating the use of autologous transplantation following ex vivo inactivation of the CCR5 gene with gene-editing (*Tebas, 2014*; *Peterson et al., 2016*). This procedure is safe and feasible in pigtail macaques infected with simian-HIV (SHIV) (*Peterson et al., 2016*; *Peterson et al., 2017*; *Peterson et al., 2018*) and is currently being investigated in a Phase I clinical trial in suppressed, HIV-1-infected humans (NCT02500849). Also, this approach is more broadly applicable because an allogeneic CCR5-negative donor is not needed. However, current data suggests that protocols do not achieve sufficient fractions of genetically modified HIV-resistant hematopoietic stem and progenitor cells (HSPCs). In contrast, in allogeneic transplant, nearly 100% of circulating immune cells after engraftment consist of donor-derived CCR5Δ32 cells. This leads to a key question: what threshold percentage of CCR5-edited, autologous HSPCs is necessary for the cure/long-term remission observed in the Berlin and London patients?

To answer this question, we developed a mathematical model that predicts the minimum threshold of gene-modified cells necessary for functional cure. First, we modeled the kinetics of CD4+-CCR5+, CD4+ CCR5-, and CD8+ T cell reconstitution after autologous transplantation. Then, we modeled SHIV kinetics during acute infection and rebound following ATI to identify the degree of loss of anti-HIV cytolytic immunity following transplantation as presented before but including some

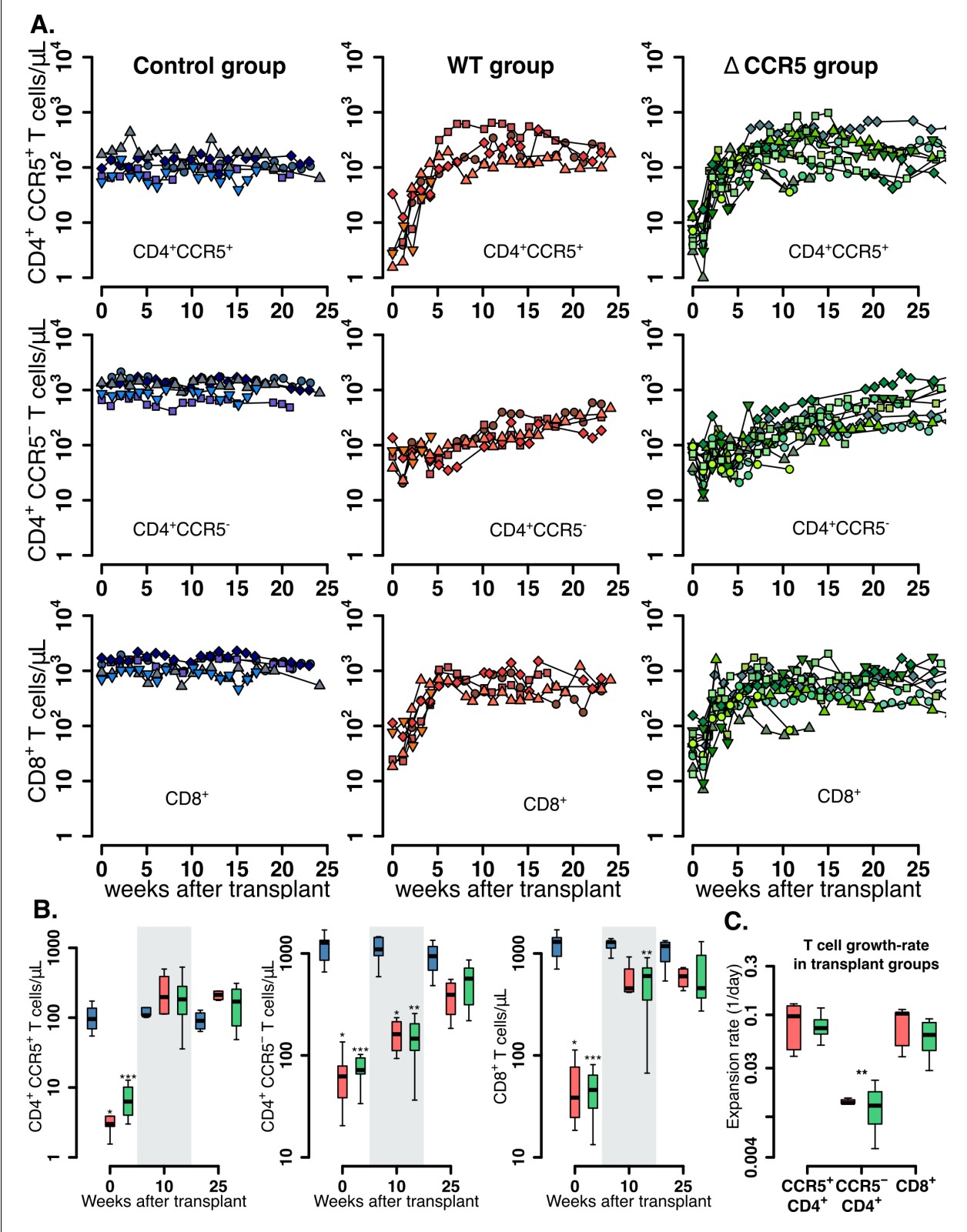

**Figure 2.** Post-transplantation, pre-ATI CD4[+] and CD8[+] T cell dynamics. (A) Empirical data for peripheral CD4[+] CCR5[+] (top row), CD4[+]CCR5[-] (middle row), and CD8[+] T cell counts (bottom row) for control (blue), wild-type (red), and ΔCCR5 (green) transplantation groups. Each data point shape and color is a different animal sampled over time. (B) Distributions of blood CD4[+] and CD8[+] T cell counts for weeks 0, 10, and 25 after transplantation (p-values calculated with pairwise Mann-Whitney test with Bonferroni correction comparing control group with transplant groups. *p<0.05, **p<0.01 and

*Figure 2 continued on next page*

*Figure 2 continued*

***p<0.001). (C) Expansion-rate estimates of CD4$^+$CCR5$^+$, CD4$^+$CCR5$^-$, and CD8$^+$ T cells (p-values calculated with paired Mann-Whitney test with Bonferroni correction comparing expansion rates of CD4$^+$CCR5$^-$ with CD4$^+$CCR5$^+$ and CD8$^+$ in transplant groups. **p<0.01 for both). Colors for boxplots in B and C are matched to A (blue: control, red: wild-type-transplantation, and green: ΔCCR5-transplantation groups).

The online version of this article includes the following source data, source code and figure supplement(s) for figure 2:

**Source code 1.** R code for plots and tests in *Figure 2*.
**Source data 1.** Complete data set of blood T cell counts for *Figures 2* and *3*.
**Figure supplement 1.** CD4$^+$ and CD8$^+$ T cell levels pre-ATI in control group (n = 5) at times relative to post-transplantation in WT and ΔCCR5 transplant groups.

additional data (*Peterson et al., 2017*; *Reeves et al., 2017*). Finally, we applied our models to predict the proportion of gene-modified cells, the dose of these cells relative to the intensity of the preparative conditioning regimen (total body irradiation, TBI), and the levels of SHIV-specific immunity required to maintain virus remission following ATI. Results from this three-part modeling approach support strategies that (1) increase stem cell dose, (2) enhance potency of conditioning regimen to reduce the number of endogenous HSPCs that compete with transplanted CCR5-edited HSPCs, (3) increase the fraction of gene-modified SHIV-resistant cells, (4) extend periods between HSPC transplantation and ATI with tracking of CCR5- cell recovery and/or (5) augment anti-HIV immunity to achieve sustained HIV remission.

## Results

### Study design and mathematical modeling

We analyzed data from 22 juvenile pig-tailed macaques that were intravenously challenged with 9500 TCID50 SHIV1157ipd3N4 (SHIV-C) (*Figure 1A*). After 6 months of infection, the macaques received combination ART that included tenofovir (PMPA), emtricitabine (FTC), and raltegravir (RAL). When on ART, 17/22 received total body irradiation (TBI) followed by the transplantation of autologous HSPCs with (n = 12) or without (n = 5) CCR5 gene editing (ΔCCR5 and WT groups, respectively). A control group (n = 5) did not receive TBI or HSPC transplantation. 14 of the animals underwent ATI approximately 1 year after ART initiation. The remaining eight animals were necropsied at an earlier time relative to the other animals' ATI (see Materials and methods for details).

To analyze the data and estimate thresholds for viral control under this approach, we used ordinary differential equation models. We performed multi-stage modeling (*Figure 1B*). First, we modeled the kinetics of CD4$^+$ and CD8$^+$ T cell subsets after autologous HSPC infusion following transplant and before ATI, assuming that ART suppression decouples SHIV-dynamics from cellular dynamics. After validation of the first-stage model, we introduced a second-stage of modeling to (1) explain virus and T cell kinetics during primary infection and ATI and to (2) identify the degree of loss of anti-HIV cytolytic immunity due to the preparative conditioning. Then, we used the final validated model to project SHIV kinetics assuming different transplantation conditions.

### CD4$^+$CCR5$^+$ and CD8$^+$ T cells recover more rapidly than CD4$^+$CCR5$^-$ T cells after HSPC transplantation

We analyzed the kinetics of peripheral blood CD4$^+$CCR5$^+$ and CD4$^+$CCR5$^-$ T cells, and total, T$_{naive}$, T$_{CM}$, and T$_{EM}$ CD8$^+$ T cells in macaques after HSPC transplantation.

In untransplanted controls, levels of CD4$^+$ and CD8$^+$ T cells oscillated around a persistent set point (blue data-points in *Figure 2A*). Also, CD4$^+$ CCR5$^+$ T cell levels were ~100 cells/μL and were uniformly lower than the CD4$^+$CCR5$^-$ T counts (each ~1000 cells/μL) (*Figure 2—figure supplement 1A*). Finally, total CD8$^+$ T cell levels in the control group were ~1400 cells/μL with a greater contribution from T$_{EM}$ (73%) than T$_N$+T$_{CM}$ (27%) (based on median values, *Figure 2—figure supplement 1*).

In the transplant groups, post-TBI levels of CD4$^+$CCR5$^+$, CD4$^+$CCR5$^-$, and CD8$^+$ T cells were significantly lower than in the control group but expanded at different rates during the following weeks (*Figure 2A–C*). The levels of CD4$^+$CCR5$^+$ T cells started at 1–10 cells/μL and reconstituted to levels similar to the control group over 5–10 weeks (*Figure 2A–B*). CD4$^+$CCR5$^-$ T cells remained at higher levels (~100 cells/μL) than CD4$^+$CCR5$^+$ T cells after TBI but expanded more slowly and did not reach

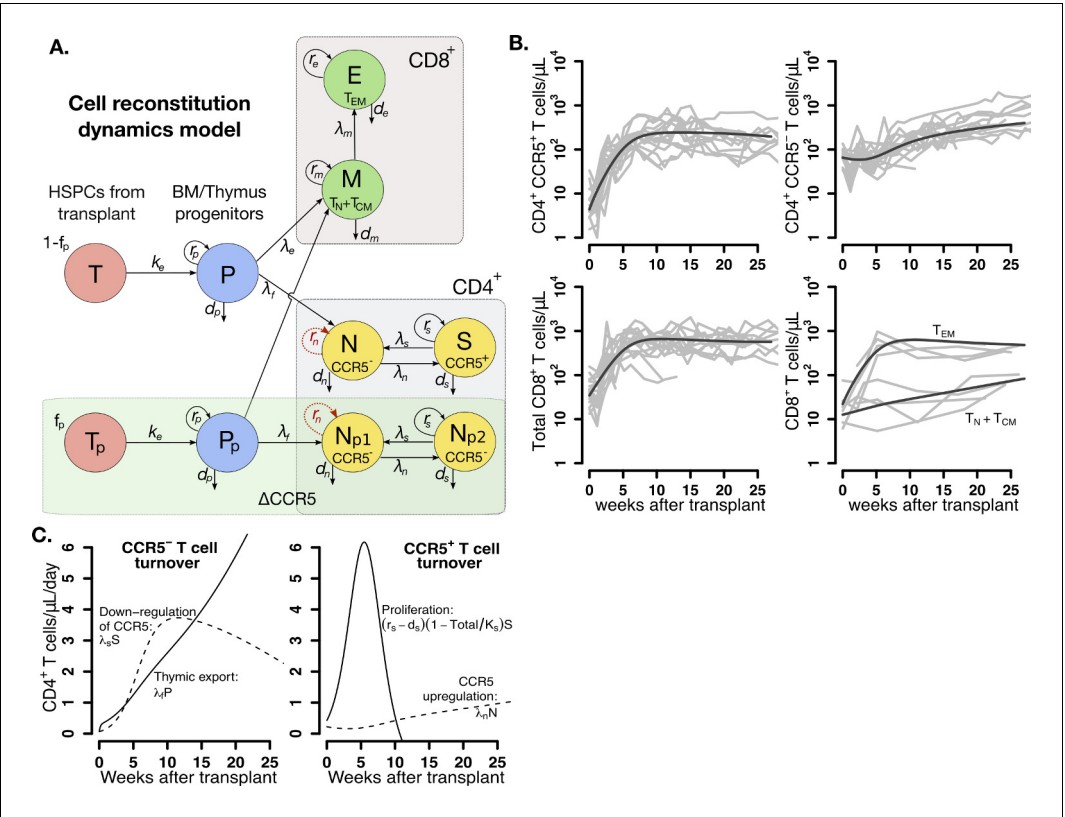

**Figure 3.** Mathematical model of T cell reconstitution after hematopoietic stem and progenitor cell (HSPC) transplantation. (**A**) Schematics of the model. Each circle represents a cell compartment: T represents the HSPCs from the transplant; P, the progenitor cells in bone marrow (BM) and thymus; S and N, CD4$^+$CCR5$^+$ and CD4$^+$CCR5$^-$ T cells, respectively; $T_p$, the protected (ΔCCR5), gene-modified cells from transplant; $P_p$, protected (ΔCCR5) progenitor cells in BM/thymus; $N_{p1}$ and $N_{p2}$ the protected (ΔCCR5) CD4$^+$ T cells; M the CD8$^+$ T cells with naive and central memory phenotype and E CD8$^+$ T cells with effector memory phenotype. The initial fraction of protected cells in the product is represented by the parameter $f_p$. Gray panels represent mature blood CD4$^+$ and CD8$^+$ T cells, and the green panel all ΔCCR5 cells in the model. Red, dashed arrows represent discarded terms after model selection and validation (see text for details). (**B**) Model predictions using the maximum likelihood estimation of the population parameters (solid black lines) for all blood T cell subsets before ATI for all animals in the transplant groups using model with ΔAIC = 0 (*Figure 3—source datas 2–3*). Each gray line is one animal. (**C**) Model predictions of the total concentration of CD4$^+$CCR5$^-$ T cells generated by CCR5 downregulation (dashed line) or thymic export (solid line), and of the total concentration of CD4$^+$CCR5$^+$ T cells generated by proliferation (solid line) or by upregulation of CCR5 (dashed line) over time using the maximum likelihood estimation of the population parameters.

The online version of this article includes the following source data, source code and figure supplement(s) for figure 3:

**Source code 1.** Best model file for T cell reconstitution in Monolix format.

**Source code 2.** R code for plots in *Figure 3*.

**Source data 1.** Values of the fraction of protected cells in transplant product $f_p$, dose or number of hematopoietic stem and progenitor cell (HSPCs) in transplant product $D$ and time of transplantation $t_x$ of each animal for model fitting and projections.

**Source data 2.** Competing models for fitting T cell reconstitution with respective AIC values.

**Source data 3.** Population parameter estimates for the best fits of the model in *Equation 2* in the main text (lowest AIC in *Figure 3—source data 2*) to the T cell reconstitution dynamics.

**Source data 4.** Individual parameter estimates for the best fits of the model in *Equation 2* in the main text (lowest AIC in *Figure 3—source data 2*) to the T cell reconstitution dynamics.

**Source data 5.** Population parameter estimates for the best fits used in the R code for *Figure 3*.

**Figure supplement 1.** Individual fits of the best model to the blood T cell observations pre-ATI in control group from a time relative to post-transplantation in transplant groups.

*Figure 3 continued on next page*

*Figure 3 continued*

**Figure supplement 2.** Individual fits of the best model to the blood T cell observations post-transplantation, pre-ATI for the wild-type-transplant group.

**Figure supplement 3.** Individual fits of the best model to the blood T cell observations post-transplantation, pre-ATI for the ΔCCR5-transplant group.

**Figure supplement 4.** Predictions of the best model for the contributors to cell expansion in CD8+ TEM cells in animals from the transplant groups.

the values of the control group after 25 weeks (*Figure 2A–B*). The CD4$^+$CCR5$^+$ T cell compartment expanded eightfold more rapidly than the CD4$^+$CCR5$^-$ compartment (p=0.008, paired Mann-Whitney test, *Figure 2C*). CD8$^+$ T cells decreased to levels between 10 and 100 cells/µL after TBI but recovered to levels just below the control group in 5 weeks (*Figure 2A–B*); CD8$^+$ T cells recovered as rapidly as the CD4$^+$CCR5$^+$ population (*Figure 2C*).

Overall, these results show that after transplantation CD4$^+$CCR5$^+$ and CD8$^+$ T cells recover faster than CD4$^+$CCR5$^-$ cells. This suggests that each cell subset may have different and/or complementary mechanisms that drive their expansion. To explore these mechanisms, we analyzed the data with a mechanistic mathematical model of cellular dynamics.

## Lymphopenia-induced proliferation drives early CD4$^+$CCR5$^+$ and CD8$^+$ T cell reconstitution after HSPC transplantation

To identify the main drivers of T cell reconstitution after transplant, we developed a mathematical model that considered plausible mechanisms underlying reconstitution of distinct T cell subsets following autologous transplantation (*Figure 3A*). We assumed that T cell reconstitution may have two main drivers: (1) lymphopenia-induced proliferation of mature cells that persist through myeloablative TBI (*Jameson, 2002*; *Schluns et al., 2002*; *Schluns et al., 2000*; *Goldrath et al., 2004*; *Voehringer et al., 2008*) and (2) differentiation from naive cells from progenitors in the thymus (from transplanted CD34$^+$ HSPCs (*Douek et al., 2000*; *Douek et al., 1998*) or residual endogenous CD34$^+$ HSPCs that persist following TBI) and further differentiation to an activated effector state (*Voehringer et al., 2008*; *Bender et al., 1999*; *Kieper and Jameson, 1999*; *Sallusto et al., 2004*; *Le Saout et al., 2008*; *Sprent and Surh, 2011*). We also assumed the infused product dose $D$ contains a fraction $f_p$ of transplanted, gene-edited HSPCs that do not express CCR5 (see *Figure 3—source data 1* for individual values of $D$ and $f_p$). Thus, in our model, ΔCCR5-gene-modified CD4$^+$ T cells differentiating from these modified HSPCs are a subset of the total CD4$^+$CCR5$^-$ cell compartment (*Figure 3A*).

We built 24 versions of the model by assuming that one or multiple mechanisms are absent, or by assuming certain mechanisms have equivalent or differing kinetics (*Figure 3—source data 2*). Using model selection theory, we identified the most parsimonious model that reproduced the data (schematic in *Figure 3A* without red-dashed lines). The best model predictions for each cell subset using maximum likelihood estimates of the population parameters (*Figure 3—source data 3*) are presented in *Figure 3B*. Individual fits are visualized in *Figure 3—figure supplement 1–3* and parameter estimates are collected in *Figure 3—source data 4*.

Model selection illuminated several likely biological phenomena: (1) CD4$^+$CCR5$^+$ T cell reconstitution after transplant is determined by cell proliferation and to a minor degree by upregulation of CCR5 (*Figure 3C*); (2) CD4$^+$CCR5$^-$ T cell expansion is driven primarily by new naive cells from the thymus and to a lesser extent by CCR5 downregulation (*Figure 3C*); and (3) thymic export is not significantly different for CD4$^+$ or CD8$^+$ T cells (*Figure 3—source data 2*). However, model selection could not distinguish between the two models where ΔCCR5-gene-modified CD4$^+$ T cells have the kinetics of both non-modified CD4$^+$CCR5$^+$ and CD4$^+$CCR5$^-$ versus only the kinetics of non-modified CD4$^+$CCR5$^-$ (i.e. having compartment $N_{p2}$ or not in *Figure 3A*). Regardless, these two best models were identical in all other respects (*Figure 3—source datas 2* and *3*).

This first-stage modeling suggested additional testable biological predictions. First, the estimated CD4$^+$CCR5$^+$ T cell proliferation rate (~0.1/day) far exceeds the estimated CCR5 upregulation (~0.004/day) and thymic export rates (~0.002/day). Therefore, 1 month after transplantation, the total concentration of CD4$^+$CCR5$^+$ T cells generated by proliferation is predicted to be 40-fold higher than the concentration generated by upregulation of CCR5 (*Figure 3C*). Second, the CD8$^+$

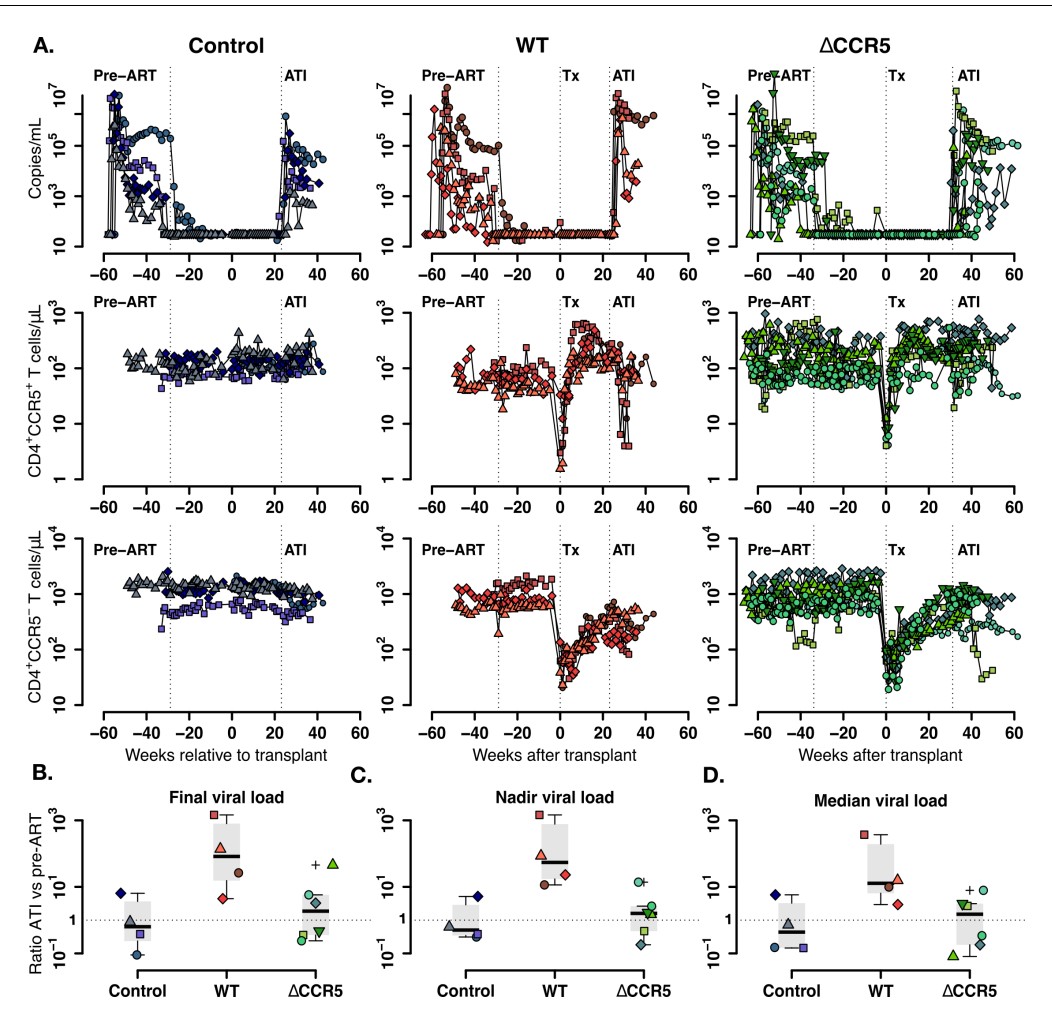

**Figure 4.** Plasma viral load and CD4$^+$ T cell kinetics after ATI. (**A**) Empirical data for viral load (top row) and peripheral T cell counts (middle and bottom rows) for control (blue), wild-type (red) and ΔCCR5 (green) transplantation groups. Each data point shape and color represent a different animal sampled over time. (**B**) Distributions of the ratio at ATI vs pre-ART of final, nadir, and median viral load. Dotted horizontal lines represent a ratio equal to one (or no difference between ATI vs nadir).

The online version of this article includes the following source data, source code and figure supplement(s) for figure 4:

**Source code 1.** R code for plots and test in *Figure 4*.

**Source data 1.** Complete data set of blood T cell counts and viral load for *Figures 4* and *5*.

**Figure supplement 1.** Blood CD4$^+$CCR5$^+$ and CD4$^+$CCR5$^-$ T cell kinetics post-ATI.

---

T$_{EM}$ cells comprise the majority of the total CD8$^+$ T cell compartment (*Figure 3B*) with a proliferation rate up to 10-fold higher than the CD8$^+$ T$_{CM}$ cell differentiation rate (*Figure 3—figure supplement 4*). In this way, CD8$^+$ T cells follow a similar pattern to CD4$^+$CCR5$^+$ T cells (*Figure 3B*).

In summary, following autologous HSPC transplant: (1) thymic export and downregulation of CCR5 drive a modest expansion of CD4$^+$CCR5$^-$ T cells, whereas (2) rapid lymphopenia-induced pro-liferation after TBI is the main driver for CD4$^+$CCR5$^+$ and CD8$^+$ T cell expansion, which are derived from both the transplanted HSPC product and residual endogenous cells that persisted through the myeloablative conditioning regimen.

## Plasma virus and blood CD4$^+$CCR5$^+$ dynamics are heterogenous among transplanted, SHIV-infected animals

To build a mathematical model for the virus and T cell dynamics, we analyzed plasma viral load kinetics and CD4$^+$CCR5$^+$/CCR5$^-$ T cell subset dynamics after ATI with respect to kinetics pre-ART (*Peterson et al., 2017*; *Reeves et al., 2017*). *Figure 4A* presents the plasma viral loads and the blood CD4$^+$CCR5$^+$ and CD4$^+$CCR5$^-$ T cell kinetics before and after transplantation in the three groups.

We calculated the ratio of the viral load at necropsy versus at initiation of ART (*Figure 4B*) and the ratio of the nadir and median viral load after ATI versus pre-ART (*Figure 4C–D*). In general, the viral burden after ATI compared to pre-ART was slightly lower for the control group. However, for transplanted animals the viral load changes were heterogeneous, having much higher ratios for the wild-type (WT) group and slightly higher for CCR5-edited (ΔCCR5) group. For the three computed ratios, the viral load change after ATI was between 10- and 100-fold for the wild-type group (*Figure 4B–D*).

During ATI, CD4$^+$CCR5$^+$ T cells declined heterogeneously in the transplanted groups (*Figure 4A*), but CD4$^+$CCR5$^+$ T cell nadirs in the transplanted groups were consistently lower than those of control animals whose CD4$^+$CCR5$^+$ T cell levels did not decrease (*Figure 4—figure supplement 1A*). On the other hand, blood CD4$^+$CCR5$^-$ T cell levels decreased to a similar nadir in all groups during ATI (*Figure 4A* and *Figure 4—figure supplement 1B*).

To summarize, SHIV viral load and CD4$^+$CCR5$^+$ dynamics are heterogeneous among transplanted animals. Higher ATI versus pre-ART viral load ratios in transplanted animals suggest that transplantation affects the host response against SHIV-replication, but this damage to host response may be mitigated somewhat when transplantation includes CCR5-edition.

## A reduction in SHIV-specific immunity leads to higher viral rebound set points following ATI in transplanted animals

We simultaneously analyzed the viral and T cell subset data using mechanistic mathematical models in order to recapitulate the heterogeneity of plasma viral load and CD4$^+$CCR5$^+$ T cell kinetics and how transplantation may modify the immune response during ATI compared to the pre-ART stage. We extended our T cell reconstitution model to include SHIV infection of CD4$^+$CCR5$^+$ T cells (*Figure 5A* and Methods) and used this second-stage model to analyze virus and T cell dynamics during primary SHIV-infection, ART, transplant, and ATI.

Again, following model selection theory based on AIC, we compared six mechanistic models and found a parsimonious model to explain the data (*Figure 5A*, *Figure 5—source data 1*). This model simultaneously recapitulates plasma viral load and the kinetics of CD4$^+$ CCR5$^+$ and CCR5$^-$ T cells as shown in *Figure 5B* and *Figure 5—figure supplements 1–3* with corresponding estimated parameters in *Figure 5—source datas 2* and *3*. In the best fitting model, parameters related to immune response against infection: the SHIV-specific CD8$^+$ T cell proliferation ($\omega_8$), saturation ($I_{50}$), and death rates ($d_h$) were different during ATI and the pre-ART stage (see *Figure 5—source data 1*; *Reeves et al., 2017*). SHIV-specific CD8$^+$ effector cells reduce virus production rather than killing infected cells (*Elemans et al., 2011*; *Klatt et al., 2010*; *Wong et al., 2010*), possibly by secretion of HIV-antiviral factors (*Shridhar et al., 2014*; *Blazek et al., 2016*; *Zhang et al., 2002*)—not explicitly included in the model. The model also suggests that infection enhances upregulation of CD4$^+$CCR5$^-$ T cells. This upregulation transiently reduces the CD4$^+$CCR5$^-$ compartment and replenishes CD4$^+$-CCR5$^+$ T cells after ATI (*Douek et al., 2003*; *Okoye et al., 2007*; *Okoye et al., 2012*). Finally, in this model, some of the ΔCCR5-gene-modified CD4$^+$ T cells also have kinetics similar to non-modified CD4$^+$CCR5$^+$ cells (i.e. it includes the compartment $N_{p2}$ as in *Figure 5A*), whereas this was not able to be differentiated in the first-stage modeling.

We used our model to compute the SHIV-specific CD8$^+$ T cell turnover rates after ATI and during pre-ART as measures of SHIV-specific immunity ($SI$) for each stage, that is, $SI_{ATI} = \frac{\omega_8^{ATI}}{d_h^{ATI}}$ and $SI_{preART} = \frac{\omega_8^{preART}}{d_h^{preART}}$, respectively. We found that the SHIV immunity ATI/pre-ART ratio $\left(\frac{SI_{ATI}}{SI_{preART}}\right)$ correlated negatively with the ATI/pre-ART ratio of the observed nadir and median viral loads (*Figure 5C–D*). In this sense, the viral burden increase during ATI (viral burden ratio >1) in animals in the transplant groups might be due to the underlying loss of the immune response to the virus ($\frac{SI_{ATI}}{SI_{preART}}$<1,

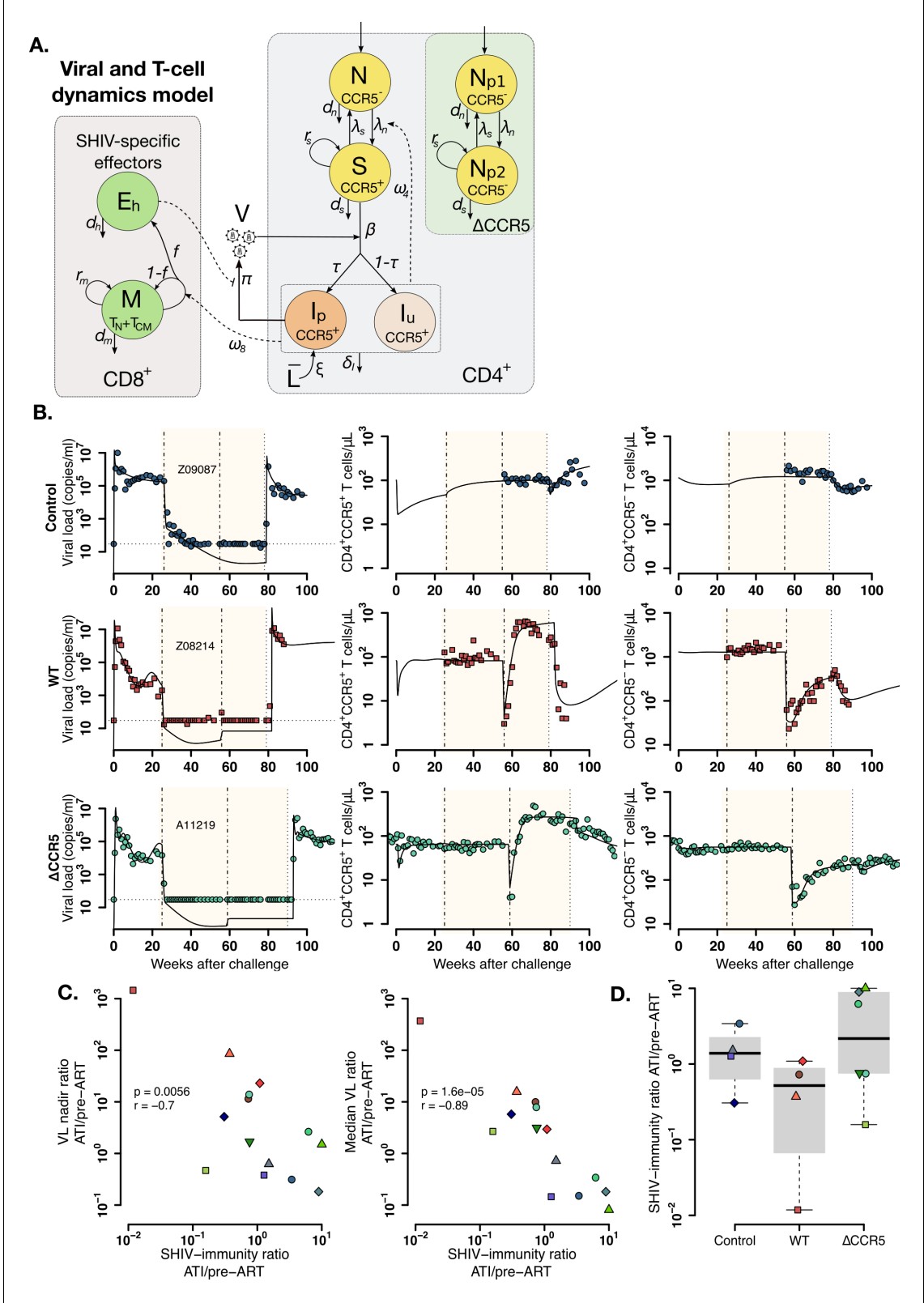

**Figure 5.** Mathematical model of virus and T cell dynamics following ATI. (**A**) Model: Susceptible cells, S, are infected by the virus, V, at rate $\beta$. $I_p$ represents the fraction $\tau$ of the infected cells that produce virus, and, $I_u$, the other fraction that becomes unproductively infected. Total $CD4^+CCR5^+$ T cell count is given by the sum of S, $I_p$ and $I_u$. All infected cells die at rate $\delta_I$. $I_P$ cells arise from activation of latently infected cells at rate $\xi\bar{L}$ and produce virus at a rate $\pi$. Virus is cleared at rate $\gamma$. $CD8^+$ M cells proliferate in the presence of infection with rate $\omega_8$ from which a fraction f become SHIV-specific

*Figure 5 continued on next page*

*Figure 5 continued*

CD8+ effector T cells, $E_h$, that are removed at a rate $d_h$. These effector cells reduce virus production ($\pi$) by 1/ (1+$\theta E_h$). Non-susceptible CD4+ T cells that were not CCR5-edited upregulate CCR5 in the presence of infection and replenish the susceptible pool at rate $\omega_4$. Gray panels represent mature blood CD4+ and CD8+ T cells, and the green panel represents ΔCCR5 cells. (B) Individual fits of the model (black lines) to SHIV RNA (left column), blood CD4+CCR5+ T cells (middle column), and CD4+CCR5- T cells (right column) for one animal in the control (top row), wild type (middle row), and ΔCCR5 groups (bottom row). Shaded areas represent time during ART and dashed-point line, the time of transplantation. (C–D) Scatterplots of observed ATI/pre-ART ratio of the (C) nadir viral load, and the median viral load ratio versus the SHIV-specific CD8+ T immunity ATI/pre-ART ratio: $\frac{\omega_8^{ATI}/d_h^{ATI}}{\omega_8^{preART}/d_h^{preART}}$ (p-values calculated by Pearson's correlation test); a higher ratio means a better immune response post-ATI. (D) Individual estimates of the SHIV-specific CD8+ T immunity ATI/preART ratio. Blue: control, red: wild type, and green: ΔCCR5 transplant group.

The online version of this article includes the following source data, source code and figure supplement(s) for figure 5:

**Source code 1.** Best model file for T cell and virus dynamics from acute infection after ATI in Monolix format.
**Source code 2.** R code for plots in *Figure 5B*.
**Source code 3.** R code for plots and tests in *Figure 5C–D*.
**Source data 1.** Competing models for fitting T cell and viral dynamics (*Equations 2-3* in main text) using the best model in *Figure 3—source data 2* and fixing parameter values as in *Figure 3—source data 3*, with AIC values.
**Source data 2.** Population parameter estimates for the fits of the model with lowest AIC in *Figure 5—source data 1* to the T cell and virus dynamics.
**Source data 3.** Individual parameter estimates for the fits of the model in *Equations 2-3* in main text (lowest AIC in *Figure 5—source data 1*) to the T cell and virus dynamics.
**Source data 4.** Individual parameter estimates obtained from Monolix for the best fits used in the R code for *Figure 5*.
**Figure supplement 1.** Individual fits of the best model to the blood T cell and viral load observations before/after ATI for control group.
**Figure supplement 2.** Individual fits of the best model to the blood T cell and viral load observations before/after ATI for the wild-type-transplant group.
**Figure supplement 3.** Individual fits of the best model to the blood T cell and viral load observations before/after ATI for the ΔCCR5-transplant group.

---

*Figure 5C–D*). Similarly, decrease in viral burden during ATI (viral burden ratio <1) in animals in control and ΔCCR5 groups might be due immune response memory or its recovery, respectively ($\frac{SI_{ATI}}{SI_{preART}}$ >1, *Figure 5C–D*).

In conclusion, we developed a second-stage model that simultaneously recapitulates viral and T cell dynamics from SHIV-infected animals receiving autologous HSPC transplantation. The model suggests that transplant may reduce host T-cell immunity resulting in higher viral loads after ATI compared to the pre-ART stage. However, SHIV immunity might be recovered if CCR5 disruption is added in the transplant resulting in lower viral loads after ATI.

## Post-ATI viral control requires a large HSPC dose containing a high fraction of CCR5-edited cells

An important advantage of our model is the ability to calculate the conditions required for post-ATI viral control (viral load set point <30 copies/ml) after CCR5-edited autologous transplant. To this end, we used our second-stage model to approximate an effective reproductive ratio $R_{eff}$ to describe the ability of the virus to sustain infection after ATI in transplanted animals (see Materials and methods):

$$R_{eff} = R_T \left(1 - \frac{f_p D}{D + P_r}\right). \tag{1}$$

Here, $f_p$ describes the fraction of protected HSPCs in the transplant product, $D$ the dose or total number of infused HSPCs, and $P_r$ the number of residual endogenous HSPCs after conditioning (variable $P$ at time of transplant, *Figure 3A*). $R_T$ is the approximate number of new infections caused by one infected cell after T cell complete reconstitution post-conditioning as defined in *Equation 4* (see Materials and methods) and is inversely related to the anti-SHIV immune response at the time of ATI. Post-ATI viral control depends on the fraction of protected HSPCs in the body immediately after transplant that are protected from SHIV infection, or $\left(\frac{f_p D}{D + P_r}\right)$.

To estimate the values of $f_p$, $D$, and $P_r$ needed for viral control, we first estimated $R_T$ for each animal based on individual parameter estimates pertaining to SHIV virulence and anti-SHIV immunity. We then simulated the model for each animal using varying values of $f_p$ from zero to one (0–100% CCR5-edited HSPCs), values of $D$ from $10^6$ to $10^9$ HSPCs, and values of $P_r$ from zero to $10^7$ HSPCs.

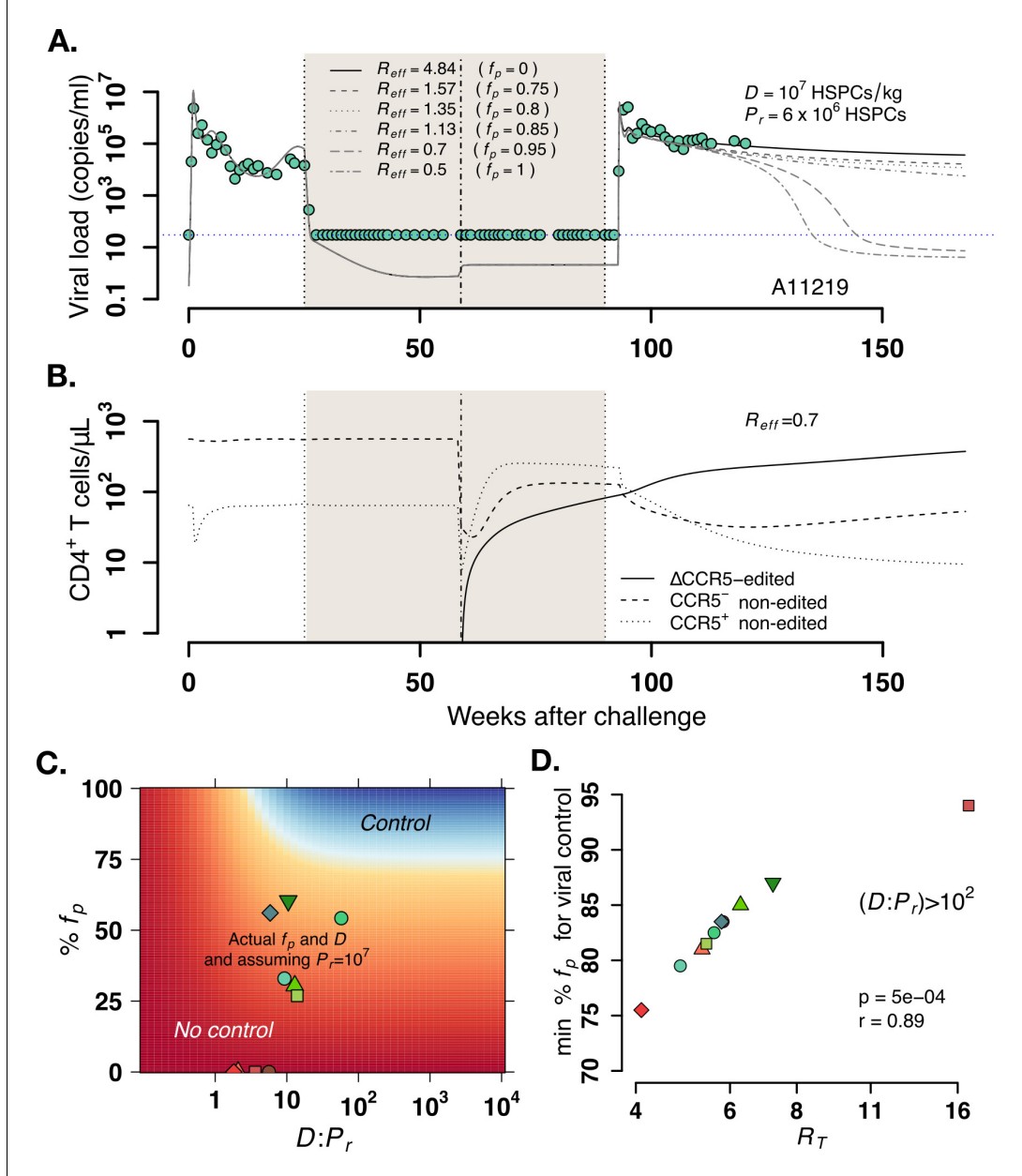

**Figure 6.** Model predictions of factors governing post-rebound viral control after CCR5 gene-edited hematopoietic stem and progenitor cell (HSPC) transplant. (A) Predictions for plasma viral loads post-ATI using the optimized mathematical model. Here, $R_{eff} = R_T \left(1 - \frac{f_p D}{D + P_r}\right)$ and is the composite determinant of viral control. Parameter estimates for animal A11219 (*Figure 5—source data 3*) were used to compute the effective reproductive ratio $R_T$. Higher values of $R_T$ imply poorer anti-SHIV immunity and high virulence (see *Equation 4* in Materials and methods). We varied values of the fraction of HSPCs in transplant $f_p$, the stem cell dose $D$ as shown, and fixed the remaining number of HSPCs after TBI before transplant $P_r = 6 \times 10^6$. $R_{eff} < 1$ predicted spontaneous viral control 40–60 weeks after ATI. (B) A simulation with $R_{eff} = 0.7$ demonstrates CCR5-edited CD4+ T cell recovery is concurrent with viral control. (C) Model predictions of the fraction of protected HSPCs in the transplant $f_p$ (y-axis) and the ratio of transplanted HSPCs to total infused plus remaining post-TBI HSPCs $D:P_r$ (x-axis) required for spontaneous viral control. The heatmap shown corresponds to animal A11200 which has $R_T = 4$, the lowest predicted $f_p$ (76%) and $D:P_r$ (~5) required for post-ATI viral control (heatmaps for other animals in *Figure 6—figure supplement 2*). Blue shaded region represents the parameter space with post-ATI viral control or $R_{eff} < 1$. Yellow-to-red region represent the parameter space with no control or $R_{eff} > 1$. Data points represent the individual values of $f_p$ and $D:P_r$ from each transplanted animal in the study. (D) Model predictions of the minimum fraction of protected HSPCs in the body $f_p$ for viral control (y-axis) for each animal given their calculated values for $R_T$ (x-axis). In all cases, the minimum $f_p$ corresponded to $\frac{D}{P_r} > 100$ (*Figure 6—figure supplement 2*). Each color is an animal, and A11200 is the red diamond with the lowest value of min $f_p$. p-Value calculated using Pearson's correlation test.

*Figure 6 continued on next page*

*Figure 6 continued*

The online version of this article includes the following source data, source code and figure supplement(s) for figure 6:

**Source code 1.** R code for plots in *Figure 6A–B*.
**Source code 2.** R code for plots in *Figure 6C–D*.
**Source data 1.** Results from all simulations varying $f_p$, $D$, and $P_r$.
**Figure supplement 1.** Model predictions for post-rebound viral control after CCR5 gene-edited hematopoietic stem and progenitor cell (HSPC) transplantation based on $R_{eff}$.
**Figure supplement 2.** Model predictions of the fraction of protected hematopoietic stem and progenitor cell (HSPCs) in the transplant $f_p$ (y-axis) and the fraction of transplanted HSPCs with respect to the total infused plus remaining post-TBI HSPCs $D:P_r$ (x-axis) required for spontaneous viral control.

As an illustration, *Figure 6A* depicts projections of the model for ΔCCR5-transplanted animal A11219 for a range of values of $f_p$ when $D = 10^7$ HSPCs/kg and $P_r = 6 \times 10^6$ HSPCs. When $f_p$, $D$ and $P_r$ resulted in $R_{eff} \geq 1$, plasma virus was not controlled following viral rebound post-ATI. When $R_{eff} < 1$, post-rebound control was observed, but only at weeks 40–60 post-ATI, following an initial decrease in viral loads beginning 30–40 weeks after ATI. In this case, post-rebound control occurred concomitantly with ΔCCR5 CD4$^+$ T cell complete reconstitution relative to non-edited CCR5$^{+/-}$ CD4$^+$ T cells (*Figure 6B*). Lower values of $R_{eff}$ resulted in earlier post-rebound control (earliest ~40 weeks).

Indeed, for all animals, post-treatment control occurred when values of $f_p$, $D$, and $P_r$ resulted in $R_{eff} < 1$ (*Figure 6—figure supplement 1*). Model predictions for animal A11200 demonstrate that regardless of the fraction of protected HSPCs in the transplant ($f_p$), viral control is possible only when the ratio of HSPCs in the transplant to the residual endogenous HSPCs in the body post-TBI ($D:P_r$) is above 5 (*Figure 6C*). Moreover, if the ratio $D:P_r$ is greater than 5, the minimum fraction of protected cells required is 76%, and further increasing $D:P_r$ does not decrease $f_p$ significantly. From all transplanted animals we found that the minimum fraction of protected cells in the transplant $f_p$ varied from 76% to 94% and was positively correlated with a weaker anti-SHIV immune response of the given animal defined by $R_T$ (*Figure 6D* and *Figure 6—figure supplement 2*). This is consistent with *Equation 1* as $R_{eff} \approx R_T (1 - f_p)$ when $D \gg P_r$. $R_T$ varied from 4 to 16 across animals using individual parameter estimates in *Figure 5—source data 3*. The required levels for $f_p$ are lower in the context of more intense anti-SHIV immunologic pressure and lower viral strength. This result argues for strategies that (1) augment anti-SHIV immunity despite conditioning (lower $R_T$ using SHIV-specific CAR T cells, therapeutic vaccination, etc.), (2) increase the stem cell dose relative to the residual endogenous stem cells ($D:P_r$) after transplant—perhaps by enhancing potency of the conditioning regimen, and (3) increase the fraction of gene-modified, SHIV-resistant cells ($f_p$).

Based on the observation that viral control occurred when CD4$^+$ T cell subsets approached a steady state in the simulations (*Figure 6A–B*), we simulated the model again to determine whether viral control might occur faster if ATI was postponed at a time when more mature, protected cells have expanded. As an illustration, we simulated animal A11219 under conditions that lead to viral control: $f_p = 0.95$, $D = 10^8$ HSPCs and $P_r = 10^7$ HSPCs with ATI occurring at 3, 14, 25, or 37 weeks after transplantation. Indeed, time to post-ATI viral control (shaded areas in *Figure 7A*) decreased as time to ATI was extended after transplant and as the difference between CD4$^+$CCR5$^-$ cell density at ATI and its expected set point decreased (shaded areas in *Figure 7B*). In this case, ΔCCR5 CD4$^+$ T cells comprised the majority of the CD4$^+$CCR5$^-$ T cell compartment (*Figure 7B*). Further, we simulated increasing times of ATI using parameter estimates for all transplanted animals but under conditions that lead to viral control ($f_p > 0.95$, $D = 10^8$ HSPCs and $P_r = 10^7$ HSPCs). The model predicted the same decreasing pattern with times between transplant and ATI required to avoid viral rebound from 20 to 60 weeks (*Figure 7C*). This timeframe allowed all animals to achieve viral control due to CD4$^+$CCR5$^-$ cell densities at ATI exceeding 60–90% of the ultimate steady state value (*Figure 7D*). As in *Figure 7B* for all animals ΔCCR5 CD4$^+$ T cells comprised the majority of the CD4$^+$CCR5$^-$ T cell compartment.

In summary, our model predicts that post-ATI viral control during autologous HSPC transplantation is obtained when (1) the transplanted HSPC dose is significantly higher than the residual endogenous HSPCs that persist through myeloablative conditioning (in this case TBI) *and* (2) the fraction of protected (i.e. CCR5-edited) HSPCs in the transplant ($f_p$) is sufficiently high to outcompete cells susceptible to infection and disrupt ongoing cycles of viral replication. Spontaneous post-rebound

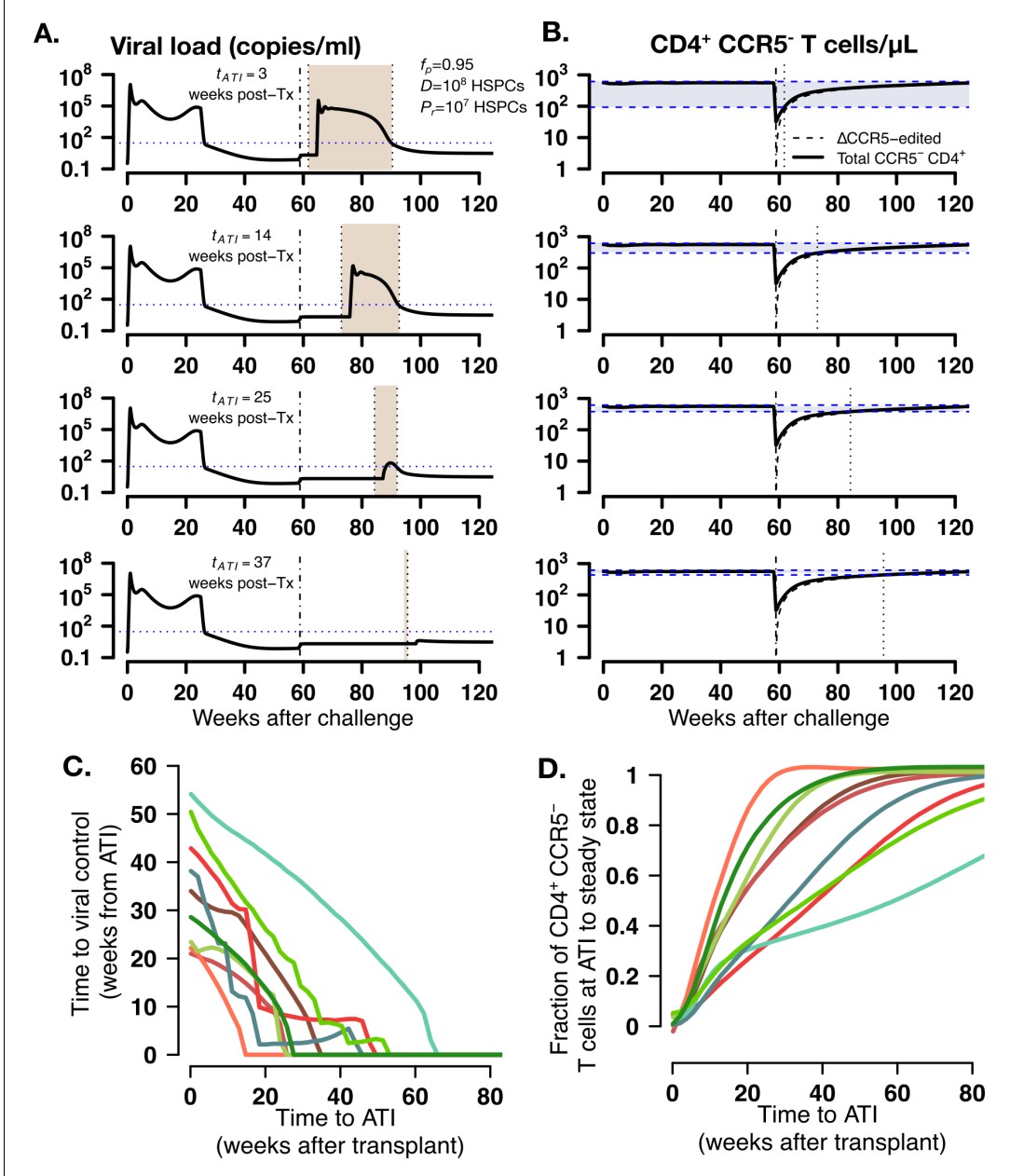

**Figure 7.** Model predictions of time to post-ATI viral control given varying times for the start of ATI. (**A-B**) Examples of projected (**A**) viral load and (**B**) total, modified and unmodified CD4$^+$ CCR5$^-$ (solid) and $\Delta$CCR5 CD4$^+$ T cells (dashed) from the model for animal A11219 when $f_p = 0.95$, $D = 10^{8.5}$ HSPCs and $P_r = 10^7$ HSPCs, for different times of ATI ($t_{ATI}$=3, 14, 25, and 37 weeks after transplantation). Dashed-dotted vertical lines represent time of transplant. Shaded areas between the dotted lines in (**A**) describe the time from ATI until spontaneous viral control. Shaded areas between the blue dashed lines in (**B**) represent the difference between the CD4$^+$ CCR5$^-$ T cell concentration at ATI and the projected steady state. Dotted lines in (**B**) represent time of ATI. (**C–D**) Model predictions of the (**C**) time until viral control after ATI and (**D**) the fraction of total CD4$^+$ CCR5$^-$ T cell concentration at ATI with respect to its steady state conditions given actual estimated parameter values for each transplanted macaque when $f_p = 0.95$, $D = 10^{8.5}$ HSPCs and $P_r = 10^7$ HSPCs.

The online version of this article includes the following source data for figure 7:

**Source data 1.** Results from all simulations varying time to ATI.

control occurs after CCR5⁻ CD4⁺ T cells achieve a steady state approximately 1-year after transplantation. Hence, our model suggests that, under the two described conditions, prolonging time to ATI (at least 1-year post-transplantation) may increase the likelihood of rapid viral control post-ATI. Moreover, specifically tracking CD4⁺CCR5⁻ (or CCR5-edited CD4⁺) T cell growth and waiting for steady-state could be used as a surrogate for the decision to undergo ATI.

## Discussion

Here we introduce a data-validated mathematical model that, to our knowledge, is the first to simultaneously recapitulate viral loads as well as CD4⁺ and CD8⁺ T cell subset counts in a macaque model of suppressed HIV-1 infection. In addition, our model is the first to describe dynamics of CCR5⁺ and CCR5⁻ T cells within the CD4 compartment. We performed extensive, systematic data fitting comparing 30 mechanistic models to arrive at a set of equations that most parsimoniously explains the available data. In multiple stages of modeling, we recapitulated (1) peripheral CD4⁺ and CD8⁺ T cell subset reconstitution dynamics following transplant and (2) T cell subset dynamics and SHIV viral rebound following ATI. Before ATI, all animals had suppressed plasma viral loads below the limit of detection, allowing analysis of T cell reconstitution dynamics independent of virus-mediated pressure. At each step, we applied model selection theory to select the simplest set of mechanisms capable of explaining the observed data (*Burnham et al., 2002*). Our model predicts that post-rebound viral control might be possible during autologous gene-edited HSPC transplantation if therapy achieves (1) a sufficient fraction of gene-protected, autologous HSPCs (2) a high dose of transplant product relative to a residual endogenous population of stem cells that persists following conditioning, and (3) enhancement of SHIV-specific immune responses following transplantation. Further, our model predicts, that under these conditions, spontaneous viral control after ATI is likely if ΔCCR5 T cells (tracked by CD4⁺CCR5⁻ T cells) are allowed to reconstitute prior to ATI. These results are consistent with the cure achieved by the Berlin and London Patients who received a transplant with 100% HIV-resistant cells after intense conditioning (*Allers et al., 2011*; *Hütter et al., 2009*). In the autologous setting where 100% CCR5 editing may not be feasible, adjunctive measures that augment virus-specific immunity, such as therapeutic vaccination, infusion of HIV-specific chimeric antigen receptor (CAR) T cells or use of neutralizing antibodies, may synergize with HSPC transplantation to achieve post-treatment control (*Haworth et al., 2017*; *Zhen et al., 2017*).

Although the model predicts a potential benefit for more potent conditioning that favors engraftment of SHIV-resistant cells, a more aggressive conditioning regimen may also deplete SHIV-specific immune responses and lead to less favorable toxicity profiles. On the other hand, in the absence of conditioning, the number of endogenous HSPCs will remain too high. Thus, post-rebound control following ΔCCR5 transplant requires not only highly potent myeloablative conditioning, it also requires a higher percentage of gene-edited cells to counteract the loss of SHIV-specific immunity. Furthermore, due to the high levels of endogenous HSPCs and lack of an engraftment 'niche', the long-term persistence of transplanted, CCR5-edited HSPC would be exceedingly low. Alternatively, non-genotoxic conditioning regimens that target only HSPC for transplantation may prevent the loss of SHIV-specific immune cells (*Palchaudhuri et al., 2016*; *Czechowicz et al., 2019*; *Srikanthan et al., 2020*).

We previously demonstrated the link between disruption of the immune response during transplant and increased magnitude of viral rebound during treatment interruption (*Peterson et al., 2017*; *Reeves et al., 2017*). Here, we confirm that the increase of viral load median and nadir at ATI compare to the pre-ART stage is correlated with the reduction of the SHIV-specific immune response during transplant, but also predict that a reduction of viral load burden at ATI compared to pre-ART in animals receiving CCR5-edited cells in the transplant is correlated to a recovery of the SHIV-specific immunity.

Our results are somewhat limited by a small sample size of 22 animals, only 12 of which underwent ΔCCR5 transplant. For that reason, several model parameters were assumed to be the same among all animals (i.e. without random effects). However, the number of observations for each animal was large enough to discriminate among several different plausible model candidates. Due to the small sample size, we also performed projections by varying the parameters related to transplantation (i.e. dose, fraction of protected cells, and residual endogenous HSPCs) and using only the estimated individual parameters rather than sampling from estimated population distributions.

Reassuringly, our results align with prior mechanistic studies of cellular reconstitution after stem cell transplantation (*Jameson, 2002*; *Douek et al., 2000*; *Guillaume et al., 1998*; *Krenger et al., 2011*; *Roux et al., 2000*). Our analysis also suggests that the majority of reconstituting CD4$^+$CCR5$^-$ T cells do not proliferate and have slow rates consistent with estimates of thymic export from previous studies (*Douek et al., 2000*; *Krenger et al., 2011*; *Roux et al., 2000*).

Recent studies from our group and others make clear that although a preparative conditioning regimen (e.g. TBI) is essential to maximize engraftment of transplanted HSPCs, it does not clear 100% of host lymphocytes, especially those in tissues (*Peterson et al., 2017*; *Peterson et al., 2018*; *Donahue et al., 2015*; *Radtke et al., 2017*). The best fitting model predicts that incomplete elimination of lymphocytes by TBI prevents CD4$^+$CCR5$^-$ cells from predominating post-transplant. We found that the rapid expansion of CD4$^+$CCR5$^+$ and CD8$^+$ T cells during the first few weeks after HSPC transplantation is most likely due to lymphopenia-induced proliferation of residual endogenous cells after TBI rather than thymic reconstitution. CD4$^+$CCR5$^-$ T cells arising from thymic export of both transplanted and remaining cells are overwhelmed by more rapidly populating CD4$^+$CCR5$^+$ T cells within weeks of transplantation. Going forward, we will need to identify anatomic sites (namely viral reservoir tissues such as spleen and lymph nodes) and associated mechanisms that allow activated CD4$^+$CCR5$^+$ to survive conditioning.

A final important observation from our model is that CD4$^+$ T cell kinetics conducive to viral control may not be reached until 20–60 weeks after transplant. Therefore, our model suggests that ATI should be delayed until CD4$^+$CCR5$^-$ T cells reconstitute (as a proxy for ΔCCR5 CD4$^+$ T cell reconstitution) to their natural steady state. Furthermore, optimized timing of ATI would ideally be based on reconstitution of all CD4$^+$ and CD8$^+$ T cell subsets ensuring approximately steady state levels before discontinuing ART.

In conclusion, our mathematical model recapitulates, to an unprecedented degree of accuracy and detail, the complex interplay between reconstituting SHIV-susceptible CD4$^+$ T cells, SHIV-resistant CD4$^+$ T cells, infected cells, virus-specific immune cells, and replicating virus following autologous, CCR5-edited HSPC transplantation. Our results illustrate the capabilities of mathematical models to glean insight from preclinical animal models and highlight that modeling will be required to optimize strategies for HIV cure.

## Materials and methods

### Study design

We employed a multi-stage approach using ordinary differential equation models of cellular and viral dynamics to analyze data from SHIV-infected pig-tailed macaques that underwent autologous HSPC transplantation during ART and to find conditions for post-rebound control when gene-edited cells were included in the transplant product. First, we modeled T cell dynamics and reconstitution following transplant and before ATI, assuming that low viral loads during suppressive ART do not affect cell dynamics (*Figure 1B*). In the second stage, we added viral load data during primary infection and after ATI and fit models to the T cell and viral dynamics simultaneously from data pre- and post-ATI (*Figure 1C*). We then used the most parsimonious model, as determined by AIC, to perform simulated experiments for different transplant conditions, focusing on variables including fraction of protected cells, dose, depletion of HSPCs after conditioning, and time of ATI after transplant to find thresholds for viral control post-ATI.

### Experimental data

Twenty-two juvenile pigtail macaques were intravenously challenged with 9500 TCID50 SHIV-1157ipd3N4 (SHIV-C) (*Peterson et al., 2017*; *Peterson et al., 2018*). After 6 months, the macaques received combination antiretroviral therapy (ART): tenofovir (PMPA), emtricitabine (FTC), and raltegravir (RAL). After ~30 weeks on ART, 17 animals received total body irradiation (TBI) followed by transplantation of autologous HSPCs. In 12/17 animals the transplant product included CCR5 gene-edited HSPCs (ΔCCR5 group); HSPC products in 5/17 animals were not edited (WT group). After an additional 25 weeks following transplant, when viral load was well suppressed, animals underwent ATI (*Peterson et al., 2017*). A control group of five animals did not receive TBI or HSPC transplantation and underwent ATI after ~50 weeks of treatment. One and six of the animals in the WT and

ΔCCR5 groups, respectively, were necropsied before ATI. One of the animals in the control group was necropsied before ATI (*Figure 1A*). Plasma viral loads and absolute peripheral T-cell counts from CD4$^+$CCR5$^-$, CD4$^+$CCR5$^+$ and total CD8$^+$ and subsets (naïve, central memory [T$_{CM}$], and effector memory [T$_{EM}$]) were measured for the control and WT group as described previously (*Peterson et al., 2017*). We analyzed peripheral T cell counts and plasma viral load from infection until 43 weeks post-transplant (~25 weeks pre-ATI and ~20 weeks post-ATI).

## Mathematical modeling of T cell reconstitution dynamics

We modeled the kinetics of CD4$^+$ and CD8$^+$ T cell subsets in blood including residual endogenous, transplanted cells that home to the BM, and progenitor cells in the BM/thymus both from transplant and residual endogenous. We included CD8$^+$ T cells in the model because CD8$^+$ and CD4$^+$ T cells may arise from new naïve cells from the thymus and compete with each other for resources that impact clonal expansion and cell survival (*Jameson, 2002*; *Mehr and Perelson, 1997*; *Margolick and Donnenberg, 1997*). We assumed that expansion of CD4$^+$ and CD8$^+$ T cells in the blood derives from: (1) export of naïve cells differentiated from a progenitor compartment in the BM/Thymus (*Guillaume et al., 1998*; *Spits, 2002*) ([either from transplanted (*Douek et al., 2000*; *Douek et al., 1998*)] or residual endogenous CD34$^+$ HSPCs) and further differentiation to an activated effector state (*Voehringer et al., 2008*; *Bender et al., 1999*; *Kieper and Jameson, 1999*; *Sallusto et al., 2004*; *Le Saout et al., 2008*; *Sprent and Surh, 2011*; *Buchholz et al., 2013*; *Farber et al., 2014*; *Kaech et al., 2002*), or (2) lymphopenia-induced division of mature, residual endogenous cells that persist through myeloablative TBI (*Jameson, 2002*; *Schluns et al., 2002*; *Schluns et al., 2000*; *Goldrath et al., 2004*; *Voehringer et al., 2008*) as factors that drive T cell proliferation are more accessible (i.e. self-MHC molecules on antigen-presenting cells [*Bender et al., 1999*; *Kieper and Jameson, 1999*; *Tanchot, 1997*] and γ-chain cytokines such as IL-7 and IL-15 [*Schluns et al., 2002*; *Schluns et al., 2000*; *Goldrath et al., 2004*; *Tan et al., 2001*]). However, as they grow, cells compete for access to these resources, limiting clonal expansion (*Jameson, 2002*) such that logistic growth models are appropriate (*Mehr and Perelson, 1997*).

In our mathematical model, transplanted HSPCs $T$ home to the bone marrow at a rate $k_e$. We assumed a single-cell compartment for T cell progenitors in the bone marrow (BM)/thymus represented by variable $P$. We assumed that $P$ renew logistically with maximum rate $r_p$, differentiate into naïve CD4$^+$ and CD8$^+$ T cells at rates $\lambda_f$ and $\lambda_e$, respectively, or are cleared at rate $d_p$ (*Stiehl and Marciniak-Czochra, 2011*; *Stiehl et al., 2014*; *Stiehl and Marciniak-Czochra, 2017*). We assumed two CD4$^+$ T cell compartments: SHIV-non-susceptible, i.e. CD4$^+$ T cells that do not express CCR5 (CD4$^+$CCR5$^-$ T cells) $N$, and a SHIV-susceptible compartment, $S$ (CD4$^+$CCR5$^+$ T cells). Only the $N$ compartment includes CD4$^+$ naïve cells migrating from the thymus (*Bleul et al., 1997*; *Zaitseva et al., 1998*; *Berkowitz et al., 1998*) at an input rate $\lambda_f P$ cells per day (*Douek et al., 2000*; *McCune, 1997*). $N$ cells grow with maximum rate $r_n$, upregulate CCR5 (27) at rate $\lambda_n$, and are cleared from the periphery at rate $d_n$. The $S$ compartment does not have a thymic input but can grow with maximum division rate $r_s$, downregulate CCR5 (27) at a rate $\lambda_s$, and are cleared at rate $d_s$. We model CD8$^+$ T cell reconstitution assuming a compartment for naïve and central memory cells, $M$, and a compartment for the effector memory subset, $E$. We assumed that $M$ cells have thymic input of $\lambda_e P$ cells per day, grow logistically with maximum division rate $r_m$, differentiate to effector memory at rate $\lambda_m$, and are cleared at rate $d_m$. The $E$ compartment grows with maximum division rate $r_e$ and is cleared at rate $d_e$. We added variables $T_p$, $P_p$, $N_{p1}$ and $N_{p2}$, representing CCR5 gene-modified- transplanted HSPCs, T cell progenitor cells in BM/thymus, and blood CD4$^+$CCR5$^-$ T cells with CD4$^+$CCR5$^-$ and CD4$^+$CCR5$^+$ kinetics, respectively. These compartments have the same structure as $T$, $P$, $N$ and $S$, but with two differences. First, the value of $T_p$ at transplantation is a fraction $f_p$ of the total number of infused cells. Second, the $N_{p1}$ cell compartment do not upregulate CCR5 when transitioning to $N_{p2}$. We model the competition of CD4$^+$ and CD8$^+$ T cells for resources that allow cell division using a logistic equation that depends on the difference between the total number of competing cells, i.e. $A = N_{p1}+N_{p2}+N+S+M+E$, and a carrying capacity $K$ (*Jameson, 2002*). Under these assumptions we constructed the following model form:

$$\frac{dT_p}{dt} = -k_e T_p$$

$$\frac{dP_p}{dt} = k_e T_p + \hat{r}_P \left(1 - \frac{A}{K_p}\right) P_p$$

$$\frac{dN_{p1}}{dt} = \lambda_f P_p + \hat{r}_n \left(1 - \frac{A}{K_n}\right) N_{p1} + \lambda_s N_{p2}$$

$$\frac{dN_{p2}}{dt} = \hat{r}_s \left(1 - \frac{A}{K_s}\right) N_{p2} + \lambda_n N_{p1}$$

$$\frac{dT}{dt} = -k_e T$$

$$\frac{dP}{dt} = k_e T + \hat{r}_p \left(1 - \frac{A}{K_p}\right) P \qquad , \qquad (2)$$

$$\frac{dN}{dt} = \lambda_f P + \hat{r}_n \left(1 - \frac{A}{K_n}\right) N + \lambda_s S$$

$$\frac{dS}{dt} = \hat{r}_s \left(1 - \frac{A}{K_s}\right) S + \lambda_n N$$

$$\frac{dM}{dt} = \lambda_e (P + P_p) + \hat{r}_m \left(1 - \frac{A}{K_m}\right) M$$

$$\frac{dE}{dt} = \lambda_m M + \hat{r}_e \left(1 - \frac{A}{k_e}\right) E$$

where $\hat{r}_p = r_p - (\lambda_f + \lambda_e + d_p)$, $\hat{r}_n = r_n - (\lambda_n + d_n)$, $\hat{r}_s = r_s - (\lambda_s + d_s)$, $\hat{r}_m = r_m - (\lambda_m + d_m)$, $\hat{r}_e = r_e - d_e$, as well as $K_w = K \frac{\hat{r}_w}{r_w}$ for each model variable $w \in \{p, n, s, m, e\}$. We did this re-parameterization to have compound parameters that were identifiable.

When simulating the model, we assumed $t_0$ as the time of transplantation. For the transplant groups the system is in a transient stage due to conditioning (TBI) at $t_0$, therefore initial values cannot be obtained from steady state equations. Transplantation is modeled as $T(t_0) = (1 - f_p)D$ and $T_p(t_0) = f_p D$. For the control group we used $t_0$ at a similar time relative to the transplant groups. Since the control group did not have any transplantation or TBI, we assumed $T(t_0) = T_p(t_0) = P_p(t_0) = N_p(t_0) = 0$. Other initial values were calculated assuming steady state: $P(t_0) = \frac{q_2 q_3 q_4 K_p}{(q_1+1)q_3 q_4 + q2(q_4+1)}$, $N(t_0) = \frac{q_1 q_3 q_4 K_p}{(q_1+1)q_3 q_4 + q2(q_4+1)}$, $S(t_0) = \frac{q_3 q_4 K_p}{(q_1+1)q_3 q_4 + q2(q_4+1)}$, $M(t_0) = \frac{q_2 q_4 K_p}{(q_1+1)q_3 q_4 + q2(q_4+1)}$ and $E(t_0) = \frac{q_2 K_p}{(q_1+1)q_3 q_4 + q2(q_4+1)}$. Here $q_1 = \frac{\hat{r}_s}{\lambda_n}\left(\frac{K_p}{K_s} - 1\right)$, $q_2 = \frac{\hat{r}_n}{\lambda_f}\left(q_1\left(\frac{K_p}{K_n} - 1\right) - \lambda_s\right)$, $q_3 = \frac{\hat{r}_m}{\lambda_e}\left(\frac{K_p}{K_m} - 1\right)$ and $q_4 = \frac{\hat{r}_e}{\lambda_m}\left(\frac{K_p}{K_e} - 1\right)$. A parsimonious, curated version of this model was selected from a series of models with varying mechanistic and statistical complexity (*Figure 3—source data 2*).

## Mathematical modeling of SHIV infection and T cell response dynamics

We next adapted the curated T cell reconstitution model by combining several adaptations of the canonical model of viral dynamics (*Reeves et al., 2017*; *Perelson, 2002*; *Perelson et al., 1997*; *Hill et al., 2018*; *Borducchi et al., 2016*; *De Boer, 2007*; *Wodarz and Nowak, 1999*; *Pandit and de Boer, 2016*). Here, virus $V$ infects only CD4$^+$CCR5$^+$ T cells (*Ho et al., 2009*) $S$ at rate $\beta$. We modeled ART by reducing the infection rate to zero. A fraction $\tau$ of the infected cells produce virus, $I_p$, and the other fraction become unproductively infected, $I_u$ (*Reeves et al., 2017*; *Doitsh et al., 2010*; *Matrajt et al., 2014*). $I_P$ cells arise only from activation of a persistent set of latently infected cells at rate $\xi \bar{L}$. We modeled ATI by assuming infection $\beta$ is greater than zero after some delay following ATI. We approximate this delay as the sum of the time of ART to washout (~3 days) and the time of successful activation ($t_{sa}$) of a steady set of latently infected cells. For simplicity, we assumed that $\xi \bar{L} = \frac{1}{t_{sa}}$ and assumed that $t_{sa}$ has lognormal distribution among the animal population with high variance (*Conway et al., 2019*; *Hill et al., 2014*; *Prague et al., 2019*). All infected cells die at rate $\delta_I$ (*Reeves et al., 2017*). $I_P$ cells produce virus at a rate $\pi$ per cell, that is cleared at rate $\gamma$. CD8$^+$ $M$ cells proliferate in the presence of infection with maximum rate $\omega_8$. A fraction $f$ of these cells become SHIV-specific CD8$^+$ effector T cells, $E_h$, that are removed at a rate $d_h$ (*De Boer, 2007*; *Wodarz and Nowak, 1999*; *Wodarz et al., 2000*). These effector cells may reduce virus production ($\pi$) or increase infected cell clearance ($\delta_I$) by 1/ (1+$\theta E_h$) or by (1+$\kappa E_h$), respectively (*Elemans et al., 2011*; *Klatt et al., 2010*; *Wong et al., 2010*; *Borducchi et al., 2016*; *Cardozo et al., 2018*). We assumed that non-susceptible CD4$^+$ T cells may upregulate CCR5 and replenish the susceptible pool during infection (*Okoye et al., 2007*; *Okoye et al., 2012*; *Okoye and Picker, 2013*) with rate $\omega_4$. For cell

growth the total number of competing cells is given by $A = N_{p1}+N_{p2}+N+S+I_p+I_u+M+E+E_h$. The model in *Equation 2* is modified to include:

$$\frac{dN}{dt} = \lambda_f P + \hat{r}_n\left(1 - \frac{A}{K_n}\right)N + \lambda_s S - \omega_4 \frac{I_p+I_u}{1+\frac{I_p+I_u}{I_{50}}}N$$

$$\frac{dS}{dt} = \hat{r}_s\left(1 - \frac{A}{K_s}\right)S + \lambda_n N - \beta VS + \omega_4 \frac{I_p+I_u}{1+\frac{I_p+I_u}{I_{50}}}N$$

$$\frac{dI_p}{dt} = \tau\beta VS - \delta_I(1+\kappa E_h)I_p + \xi\bar{L}$$

$$\frac{dI_u}{dt} = (1-\tau)\beta VS - \delta_I(1+\kappa E_h)I_u$$

$$\frac{dV}{dt} = \frac{1}{1+\theta E_h}\pi I_p - \gamma V$$

$$\frac{dM}{dt} = \lambda_e(P+P_p) + \hat{r}_m\left(1 - \frac{A}{K_m}\right)M + \omega_8(1-2f)\frac{I_p+I_u}{1+\frac{I_p+I_u}{I_{50}}}M$$

$$\frac{dE_h}{dt} = \omega_8 f\frac{I_p+I_u}{1+\frac{I_p+I_u}{I_{50}}}M - d_h E_h.$$

$$(3)$$

When simulating this model, we assume $t_0 = 0$ as the moment of SHIV challenge, and $t_x$ as the moment of transplantation after challenge. We modeled conditioning by: (1) adding a term $-k_T C$ in all blood cell compartments $C \in \{N, S, I_p, I_u, M, E, E_h\}$ and (2) the term $-k_H P$ for the HSPC compartment $P$. $k_T$ and $k_H$ are different than zero only during the 2 days before transplant ($t_x - 2 \leq t < t_x$). Transplantation is modeled as an input only when $t = t_x$ to cell compartments $T$ and $T_p$ with amounts $(1-f_p)D$ and $f_p D$, respectively. A parsimonious version of this model was selected from a series of models with varying mechanistic and statistical complexity (*Figure 3—source data 2*).

## Nonlinear mixed-effects modeling

To fit our models (*Equations 2, 3*) to the transplant data, we used a nonlinear mixed-effects modeling approach (*Lavielle, 2014*). Within this approach, we modeled a state variable vector $v$ with observations at time $i$ for each animal $j$ as $log_{10}v_{ij} = f_v(t_{ij}, \Psi_j) + \epsilon_v$. Here, $f_v$ describes the solution of the nonlinear models in *Equations 2* or 3 for the state variable vector $v$ at observation time $t_{ij}$ with animal-specific parameter set $\Psi_j$. The distribution of measurement noise is assumed as $\epsilon_v \sim \mathcal{N}(0, \sigma_v^2)$.

In the mixed-effects model, it is assumed that for an animal $j$ each single parameter $\psi_j \in \Psi_j$ is drawn from a probability distribution across the population. This distribution includes the fixed effects $\bar{\psi}$ representing the median value over the population, and the random effects $\eta_j$ representing its variability in the population, assumed to be normally distributed with standard deviation $\sigma_\psi$, that is $\eta_j \sim \mathcal{N}(0, \sigma_\psi^2)$. We assumed that the random effects of the parameters $\eta_j$ might not be independent. In that case, the vector of random effects $\eta_j$ follows a multinormal distribution: $\eta \sim \mathcal{N}(0, \Omega)$, being $\Omega$ the variance-covariance matrix based on the values $\sigma_\psi$ and correlations between the individual parameters in $\eta$.

We fit each model to all data points from all animals simultaneously using a maximum likelihood approach. We assumed that individual observations of each state variable $v_{ij}$ for each animal $j$ at each time point $t_{ij}$ are independent. For each model, we obtained the Maximum Likelihood Estimation (MLE) of the standard deviation of the measurement error for the observations $\sigma_v$, and each parameter fixed effects $\bar{\psi}$ and standard deviation of the random effects $\sigma_\psi$ (or elements in matrix $\Omega$ when applicable) using the Stochastic Approximation of the Expectation Maximization (SAEM) algorithm embedded in the Monolix software (http://www.lixoft.eu).

## Fitting T cell reconstitution before ATI

We first fit the observed blood T cell kinetics after HSPC transplantation and before analytical treatment interruption (ATI) using the model in *Equation 2*. During this procedure, we defined the vector $v^{(1)}$ to model the $log_{10}$ of the observed blood CD4$^+$CCR5$^-$, CD4$^+$CCR5$^+$, total CD8$^+$, CD8$^+$ T$_N$ + T$_{CM}$, and CD8$^+$ T$_{EM}$ cell counts which are represented in *Equation 2* by the variables $\{N + N_{p1} + N_{p2}, S, C, M, E\}$, respectively with $C = M + E$ and solution $f^{(1)}$.

We defined the statistical form of each parameter in $\Psi^{(1)}$ in the following form: parameters $r_p^j, r_m^j, r_e^j, \lambda_f^j, \lambda_e^j, \lambda_n^j, \lambda_s^j, \lambda_m^j$ were modeled as $\psi_j = \bar{\psi}e^{n_j}$; parameter $K_p^j$ was modeled as $\psi_j = 10^{\bar{\psi}+n_j}$; $K_n^j, K_s^j, K_m^j, K_e^j$ were modeled as $\psi_j = 10^{K_p^j - \bar{\psi}e^{n_j}}$; and initial values in the transplant group:

$N^j(t_0), S^j(t_0), M^j(t_0)$ and $E^j(t_0)$ had the model $\psi_j = 10^{\bar{\psi}+n_j}$. We explored the possibility that $r_n = 0$, in that case we assumed $\hat{d}_n^j = \lambda_n^j(1 + \bar{\psi}e^{n_j})$. We fixed the HSPC homing rate $k_e$ = 1/day (*Lapidot et al., 2005*; *Chute, 2006*), and $f_p$ and $D$ as described in *Figure 3—source data 1*. Since at $t_0$ the system is in a transient stage due to conditioning (TBI), we estimated blood cell concentrations at $t_0$, but fixed the number of HSPCs that remained in the BM/thymus $P(t_0)$ to $6 \times 10^6$ based on the estimated minimum number of infused HSPCs needed for engraftment in the same animal model (*Radtke et al., 2017*).

We fit instances of models with varying statistical and mechanistic complexity in *Equation 2* to blood T cell counts during transplant and before ATI (*Figure 1B*) assuming that one or multiple mechanisms are absent, or that certain mechanisms have equal kinetics (*Figure 3—source data 2* includes all 24 competing models with the different statistical assumptions).

## Fitting T cell and viral load dynamics before and after ATI

Next, we fit the model in *Equations 2-3* to the pre- and post-ATI blood T cell counts and plasma viral loads (*Figure 1B*). Here, we defined $v^{(2)}$ for variables $\{N + N_{p1} + N_{p2}, R, C_4, C_8, M, E, V\}$ with $V$ indicating the observed plasma viral load, $N + N_{p1} + N_{p2}$ indicating the observed blood CD4$^+$CCR5$^-$ T cell concentration, $R = S + I_p + I_u$ the observed blood CD4$^+$CCR5$^+$ T cell concentration, $C_8$ the total CD8$^+$ T cell concentration, $C_4 = R + N + N_{p1} + N_{p2}$ the total CD4$^+$ T cell concentration and the others state-variables as specified for $v^{(1)}$. We included $C_4$ because we had total CD4$^+$ T cell counts, but CD4$^+$ T subset counts during the primary infection stage were not available in many of the animals. For this model, we defined the parameter set $\Psi^{(2)}$ by adding to the parameters in the previous section the parameters relative to virus dynamics (i.e. $\Psi^{(2)} = \{\Psi^{(1)}, \kappa^j, \theta^j, \beta^j, \pi^j, \omega_4^j, \omega_8^j, I_{50}^j, d_h^j, t_{sa}^j\}$ but fixing the values in $\Psi^{(1)}$ to the MLE values using *Figure 3—source data 3*). For parameters $\kappa^j, \theta^j, \beta^j, \pi^j, \omega_4^j, \omega_8^j, I_{50}^j$, we used a model with form $\psi_j = 10^{\bar{\psi}+n_j}$, and for $d_h^j$ and $t_{sa}^j$ we used $\psi_j = \bar{\psi}e^{n_j}$. We included the possibility that immunity might be different at ATI compared to pre-ART by assuming the forms $\psi^{j,ATI} = 10^{\bar{\psi}+n_j+\varsigma_{\psi,ATI}}$ for $\omega_8^j$ and $I_{50}^j$, and $\psi^{j,ATI} = \bar{\psi}e^{n_j+\varsigma_{\psi,ATI}}$ for $d_h^j$ during ATI. We evaluated single or combination of mechanistic hypotheses along with different statistical assumptions as listed in *Figure 5—source data 1* using AIC. $V(0)$ was fixed to a small value below the limit of detection, and $I_p(0)$ and $I_u(0)$ were calculated as $\tau c V(0)/\pi$ and $(1 - \tau)c V(0)/\pi$, respectively. We fixed the following parameters: $\gamma = 23$/day (*Ramratnam et al., 1999*), $\delta_I = 1$/day (*Markowitz et al., 2003*; *Cardozo et al., 2017*), $\tau = 0.05$ (*Doitsh et al., 2010*), and $f = 0.9$ (*Borducchi et al., 2016*). The value of $k_h$ was constrained to obtain a value of the HSPCs after conditioning $P(t_x) = P_r = 6 \times 10^6$ (*Radtke et al., 2017*). We fixed values of $t_x, f_p$ and $D$ as described in *Figure 3—source data 1*.

We fit several instances of the model in *Equation 3* to pre- and post-ATI blood T cell counts and plasma viral loads (*Figure 1B*) using the best model obtained for *Equation 2* (*Figure 5—source data 1* includes all four competing models and respective statistical assumptions). At the time of SHIV infection, values for the cell compartments were calculated from steady state equations with the same form as for the group without transplantation ('control') in the previous section.

## Model selection

To determine the best and most parsimonious model among the instances, we computed the log-likelihood (log $L$) and the Akaike Information Criteria (AIC=-2log $L$+2 m, where $m$ is the number of parameters estimated) (*Burnham et al., 2002*). We assumed a model has similar support from the data if the difference between its AIC and the best model (lowest) AIC is less than two (*Burnham et al., 2002*).

## Effective reproductive ratio when $r_n = 0$ and $\kappa = 0$

We calculated an approximate effective reproductive ratio $R_{eff}$ for our model (*Equations 2, 3*) by computing the average number of offspring produced by one productively infected cell $I_p$ at ATI assuming all cell compartments have reached steady state after transplantation during ART. This number is the product of the average lifespan of one $I_p$, the virus production rate by this latently infected cell, the lifespan of produced virions from this cell, the rate at which each virion infects the

pool of susceptible cells at steady state, the fraction of these infections that become productive and the reduction of virus production, cell infection, and cell death by SHIV-specific immune cells at ATI. Using this approach, we obtain that $R_{eff} = \frac{\tau \beta \bar{S} \pi}{\gamma \delta_I (1 + \theta \bar{E}_h)}$, with $\bar{S} \approx \frac{\lambda_f \bar{P}}{\frac{\hat{d}_n \hat{r}_s}{\lambda_n}(\frac{K_p}{K_s}-1) - \lambda_s}$ and $\bar{E}_h \approx \frac{f \omega_8 \lambda_e K_p}{a d_h \delta_I t_{sa} \hat{r}_m (\frac{K_p}{K_m}-1)}$ the steady state values of variables $S$ (SHIV-susceptible cells) and $E_h$ (SHIV-specific effector cells) during ART, with $a = \frac{\lambda_e}{\hat{r}_m(\frac{K_p}{K_m}-1)} + \frac{\lambda_f \lambda_n}{\hat{d}_n \hat{r}_s(\frac{K_p}{K_s}-1) - \lambda_s \lambda_n} + \frac{\lambda_f \hat{r}_s (\frac{K_p}{K_s}-1)}{\hat{d}_n \hat{r}_s(\frac{K_p}{K_s}-1) - \lambda_s \lambda_n} + \frac{\lambda_e \lambda_m}{\hat{r}_m(\frac{K_p}{K_m}-1)\hat{r}_e(\frac{K_p}{K_e}-1)} + \frac{f \omega_8 \lambda_e}{d_h \delta_I t_{sa} \hat{r}_m (\frac{K_p}{K_m}-1)}$. By assuming that the total amount of infused cells (dose $D$ and fraction of CCR5-editing $f_p$) home to the BM/Thymus rapidly, and that the amount of remaining HSPCs after TBI and immediately before transplant is $P(t_x) = P_r$, the approximate steady state for $P$ is $\bar{P} \approx \frac{K_p}{a} \cdot \frac{(1-f_p)D+P_r}{D+P_r} = \frac{K_p}{a}\left(1 - \frac{f_p D}{D+P_r}\right)$. Together this gives the following expression for the effective reproductive ratio:

$$R_{eff} = R_T\left(1 - \frac{f_p D}{D+P_r}\right), \text{ with } R_T = \frac{\tau \beta \pi \lambda_e K_p}{a \gamma \delta_I \left[\frac{\hat{d}_n \hat{r}_s}{\lambda_n}\left(\frac{K_p}{K_s}-1\right) - \lambda_s\right](1 + \theta \bar{E}_h)} \tag{4}$$

Here, $R_T$ then represents the effective reproductive ratio during transplant in the absence of gene-editing when cells have reached steady state.

## Acknowledgements

This study was supported by grants from the National Institutes of Health, National Institute of Allergy and Infectious Diseases (UM1 AI126623, R01 AI150500). ERD is supported by the National Center for Advancing Translational Sciences of the National Institutes of Health under Award Number KL2 TR002317. DBR is supported by a Washington Research Foundation postdoctoral fellowship, and a CFAR NIA P30 AI027757. NHP studies were supported by NIH P51 OD010425. The funders had no role in study design, data collection and analysis, decision to publish, or preparation of the manuscript. The content is solely the responsibility of the authors and does not necessarily represent the official views of the National Institutes of Health or the Washington Research Foundation.

## Additional information

### Competing interests
Hans-Peter Kiem: H.-P.K has served on advisory boards for Rocket Pharmaceuticals, Homology medicines and CSL for research unrelated to this manuscript. The other authors declare that no competing interests exist.

### Funding

| Funder | Grant reference number | Author |
| --- | --- | --- |
| National Institute of Allergy and Infectious Diseases | UM1 AI126623 | E Fabian Cardozo-Ojeda<br>Christopher W Peterson<br>Hans-Peter Kiem<br>Joshua T Schiffer |
| National Institute of Allergy and Infectious Diseases | R01 AI150500 | E Fabian Cardozo-Ojeda<br>Joshua T Schiffer |
| National Center for Advancing Translational Sciences | KL2 TR002317 | Elizabeth R Duke |
| Center for AIDS Research | New Investigator Award P30 AI027757 | Daniel B Reeves |
| Washington Research Foundation | Postdoctoral Fellowship | Daniel B Reeves |
| National Institutes of Health | P51 OD010425 | Hans-Peter Kiem |

The funders had no role in study design, data collection and interpretation, or the decision to submit the work for publication.

## Author contributions
E Fabian Cardozo-Ojeda, Software, Formal analysis, Funding acquisition, Validation, Investigation, Visualization, Methodology, Writing - original draft, Writing - review and editing, Developed the mathematical models, wrote all code, performed all calculations and derivations, ran the models in Monolix and analyzed output data; Elizabeth R Duke, Visualization, Methodology, Writing - review and editing, Development of mechanistic mathematical models; Christopher W Peterson, Data curation, Writing - review and editing; Daniel B Reeves, Methodology, Writing - review and editing, Development of mechanistic mathematical models; Bryan T Mayer, Methodology, Writing - review and editing, Contributed ideas and support for statistical models and analyses; Hans-Peter Kiem, Data curation, Funding acquisition, Writing - review and editing; Joshua T Schiffer, Conceptualization, Supervision, Funding acquisition, Methodology, Writing - original draft, Writing - review and editing

## Author ORCIDs
E Fabian Cardozo-Ojeda (iD) https://orcid.org/0000-0001-8690-9896
Daniel B Reeves (iD) http://orcid.org/0000-0001-5684-9538
Joshua T Schiffer (iD) https://orcid.org/0000-0002-2598-1621

## Ethics
Animal experimentation: The data used in this work were collected in strict accordance with the recommendations in the Guide for the Care and Use of Laboratory Animals of the National Institutes of Health. The study protocol was approved by the Institutional Animal Care and Use Committees (IACUC) protocols (#3235-03) of the Fred Hutchinson Cancer Research Center and the University of Washington.

## Decision letter and Author response
Decision letter https://doi.org/10.7554/eLife.57646.sa1
Author response https://doi.org/10.7554/eLife.57646.sa2

## Additional files
### Supplementary files
• Transparent reporting form

### Data availability
All data generated or analysed during this study are included in the manuscript and supporting files. Source data files have been provided for Figures 2 to 7. Details of the source data for each figure are in the Transparent Reporting form.

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
