## [Decision Letter]

**Acceptance summary:**

The search for an HIV 'cure' includes a number of approaches. These include eradication of the latent virus, as well as strategies to prevent viral replication. The known clinical examples of HIV 'cure' have occurred in patients that have received bone marrow transplants where the new donor marrow lacked the CCR5 gene that encodes an HIV co-receptor molecule. Thus, understanding how reduced CCR5 expression affects viral replication is an important question. In this study, the authors investigate viral control in SHIV infected macaques following autologous CCR5 gene-edited transplantation. To this end, they combine experimental data on SHIV-infected macaques with mathematical models describing viral and immune cell dynamics using a stepwise approach. This as a very innovative interdisciplinary study that provides important insights to inform potential future treatment regimens against HIV.

**Decision letter after peer review:**

Thank you for submitting your article "Thresholds for post-rebound SHIV control after CCR5 gene-edited autologous hematopoietic cell transplantation" for consideration by *eLife*. Your article has been reviewed by Aleksandra Walczak as the Senior Editor, a Reviewing Editor, and two reviewers. The reviewers have opted to remain anonymous.

The reviewers have discussed the reviews with one another and the Reviewing Editor has drafted this decision to help you prepare a revised submission.

As the editors have judged that your manuscript is of interest, but as described below that additional work is required before it is published, we would like to draw your attention to changes in our revision policy that we have made in response to COVID-19 (https://elifesciences.org/articles/57162). First, because many researchers have temporarily lost access to the labs, we will give authors as much time as they need to submit revised manuscripts. We are also offering, if you choose, to post the manuscript to bioRxiv (if it is not already there) along with this decision letter and a formal designation that the manuscript is "in revision at *eLife*". Please let us know if you would like to pursue this option. (If your work is more suitable for medRxiv, you will need to post the preprint yourself, as the mechanisms for us to do so are still in development.)

Summary:

In this study, the authors investigate the requirements for viral control in SHIV infected macaques under CCR5 gene-edited HSPC transplantation treatment. To this end, they combine experimental data on SHIV-infected macaques with mathematical models describing viral and immune cell dynamics using a stepwise approach. The Materials and methods and Results are presented in a thorough way. The article utilizes a robust data set and also uses detailed mathematical models and extensive analyses to capture the relevant dynamics.

Essential revisions:

1) The authors perform model selection based on the AIC values to determine the appropriate mathematical model describing the observed dynamics. However, the reliability of the obtained AIC values also depends on the identifiability of the estimated model parameters. Can you say anything about the extent to which you are overfitting in these models? The number of parameters is quite large and you see some quite substantial correlations (Supplementary file 3).

2) Conditioning and CCR5-deleted autologous HSCT depletes existing infected cells and pre-existing immunity to SHIV and ultimately increases the proportion of infection-resistant cells after reconstitution. It's therefore not obvious that extensive conditioning is necessarily the best strategy, and you have shown this in Peterson et al., 2017 and Reeves et al., 2017. Could this be worth highlighting early on? It may help to increase the impact of the insights you derive from the modeling.

3) When modeling reconstitution in deltaCCR5 individuals – you assume that the CCR5- cells all behave like CCR5- cells in WT transplants. This doesn't seem like a valid assumption; a proportion of CCR5-/- cells will become activated (as they do in WT transplants) and so the CCR5-/- cell kinetics should be a mixture of the CCR5- and CCR5+ kinetics. The same parameters could be used in the two groups. How sensitive are your conclusions to this issue?

4) Subsection “A reduction in SHIV-specific immunity leads to higher viral rebound set points and CD4^+^CCR5+ T cell depletion following ATI in transplanted animals” and Figure 5E – this section needs more interpretation/discussion. It's surprising that the only difference in parameters between the control and treatment groups was the time to virus rebound; so presumably differences across groups derive from the steady state sizes of the different populations? Are latently infected cells depleted in the transplant groups? And it's puzzling that delayed rebound in transplant groups is taken to imply more depleted immunity. Shouldn't it imply less depletion? Also – elevated virus load after ATI only occurs in the WT transplant group, not both. Please clarify and expand this section.

5) Subsection “Post-ATI viral control requires a large HSPC dose containing a high fraction of CCR5-edited cells”: Are strategies 1 and 2 not in conflict? Or does more potent conditioning somehow not necessarily imply a greater reduction in anti-SHIV immunity?

In summary, the reviewers acknowledged the analysis of an important topic by a unique dataset. However, they identified some major aspects that addressed the parameter identifiability within the model estimates as well as identifying the role of conditioning and dynamics of CCR5- cells. Therefore, we would like to ask you to analyze/comment on parameter identifiability for individual model estimates and validate the robustness of parameter estimates and, hence, model selection. This will be subject to re-review to determine if these issues have been satisfactorily addressed.

[Editors' note: further revisions were suggested prior to acceptance, as described below.]

Thank you for resubmitting your work entitled "Thresholds for post-rebound SHIV control after CCR5 gene-edited autologous hematopoietic cell transplantation" for further consideration by *eLife*. Your revised article has been evaluated by Aleksandra Walczak (Senior Editor) and a Reviewing Editor.

The manuscript has been improved but the reviewers have identified some remaining issues that need to be addressed before acceptance, as outlined below:

1) Figure 2—figure supplement 1, Panel B: The statistical comparison total vs. TN+TCM does not make sense. As I assume that Total=T_N_+T_CM_+T_EM_, "Total" does not contain additional information when there is already a comparison T_N_+T_CM_ vs. T_EM_.

2) The manuscript contains a lot of typos and needs proofreading.

3) The Berlin and London patients are still not described.

4) The authors should provide all data and code used to perform the analyses.

---

## [Author Response]

Essential revisions:1) The authors perform model selection based on the AIC values to determine the appropriate mathematical model describing the observed dynamics. However, the reliability of the obtained AIC values also depends on the identifiability of the estimated model parameters. Can you say anything about the extent to which you are overfitting in these models? The number of parameters is quite large and you see some quite substantial correlations (Supplementary file 3).

We carefully took into account the following factors to reduce overfitting of the model and unidentifiability of the parameters as much as possible:

1) We divided the fits in two steps: (1) T cell reconstitution following TBI during virus suppression while on ART and (2) T cell and virus dynamics for the whole study period. The first step modeling fits to the isolated kinetics of T cells: expansion, overshoot and steady state. This method allows identification of parameters related to T cell reconstitution that do not correlate with parameters of virus dynamics. In the second step we fit to the same T cell subsets and to viral load data from acute infection and during ART interruption, fixing the parameters unrelated to virus infection and response from the first step. By using the two steps, parameters of the T cell dynamics related and unrelated to the virus dynamics are separately estimated.

2) The data during step (1) was frequently sampled for multiple T cell subsets allowing parameter identification related to each subset that were unrelated to virus dynamics (which is normally difficult when fitting models to viral loads only).

3) We transformed the model for step (1) as presented in the Materials and methods section after Equation 2 to ensure structural identifiability. Specifically, we combined several parameters, such as proliferation (r_i_), transition/differentiation and death rates (d_i_) into compound parameters and estimated the compound parameters. Whereas the individual rates were not identifiable, compound rates (capturing correlated behaviors) were.

4) The reparameterization shows that some different differentiation parameters (λ_i_) cannot be equal to zero, but others can be. For those that could take on a value of zero, we performed model selection to obtain a better fit and avoid overfitting.

5) We did careful model selection with reasonable models for step (1) (Figure 3—source data 2) using AIC to reject the most complex models and thus avoid overfitting (given the large amount of data compared to the number of parameters, AICc did not result in different outcomes). For each model we were careful in evaluating which parameters might not be practically identifiable and which ones correlated using the software Monolix:

a) For practical identifiability we used the relative standard error (standard error/population parameter value). Since the standard error is based on the diagonal of the inverse of the Fisher information matrix, it describes how much information each parameter gives to calculate the likelihood, and therefore gives information on how identifiable a parameter can be relative to the available data. We made sure the best model had small % relative standard error (%RSE<=40) for most of the estimated parameters (Figure 3—source data 3). For some parameters, the standard deviation of the random effects were estimated to be close to zero but with very high %RSE values. We therefore assume those parameters have no random effects (standard deviation equal to zero), i.e. all animals have the same estimated individual value for that parameter.

b) For correlations among parameters Monolix also uses the inverse of the Fisher Information Matrix to give information about those correlations. For those with high correlation we fit the model again but explicitly including the statistical relationship between the parameters in the variance-covariance matrix Ω of the mixed-effects model (corr≠0 in Figure 3—source data 2 and Figure 5—source data 1 means that the correlations were explicitly part of Ω in the mixed-effect model—therefore they were estimated parameters also included in the AIC). We made sure that for the best model the estimated correlation was identifiable (i.e. low %RSE in Figure 3—source data 3 and Figure 5—source data 2). This means that when drawing individual parameter values from the distribution in the mixed-effect model, the estimated correlation was also taken into account.

6) Because of the possibility of lack of convergence by the SAEM fitting algorithm in Monolix, we repeated the fitting procedure for each model 10 times having randomly selected initial guesses for the parameters. From the 10 attempts we selected the case with the highest likelihood.

7) When fitting the model for step (2) we fixed all the estimated parameters in step one, including those with correlations. For practical identifiability and correlations of the parameters related with virus dynamics and response to the virus we followed the same procedure as in item 5 (all summarized in the “statistical assumptions” in Figure 5—source data 1. See also the %RSE for best model estimates in Figure 5—source data 2 – most of them with %RSE <=40%).

8) For both steps, we fixed some parameters based on estimation from previous studies.

We recognized the possibility of model overfitting and unidentifiability of parameters, but we believe that the careful steps we took reduced those two problems significantly.

2) Conditioning and CCR5-deleted autologous HSCT depletes existing infected cells and pre-existing immunity to SHIV and ultimately increases the proportion of infection-resistant cells after reconstitution. It's therefore not obvious that extensive conditioning is necessarily the best strategy, and you have shown this in Peterson et al., 2017 and Reeves et al., 2017. Could this be worth highlighting early on? It may help to increase the impact of the insights you derive from the modeling.

We have included a sentence in the Introduction explaining that extensive conditioning is a necessary condition (despite the loss of pre-existing immunity) to reduce the number of endogenous HSCs so that transplanted CCR5-deleted HSCs can outcompete with them.

We also added a sentence in the Discussion saying that in the absence of conditioning the number of endogenous HSPCs will remain too high. In this case, the amount that would have to be taken from the animals for ex-vivo CCR5 deletion and transplantation for SHIV remission would be unrealistic, and their engraftment potential would be low.

3) When modeling reconstitution in deltaCCR5 individuals – you assume that the CCR5- cells all behave like CCR5- cells in WT transplants. This doesn't seem like a valid assumption; a proportion of CCR5-/- cells will become activated (as they do in WT transplants) and so the CCR5-/- cell kinetics should be a mixture of the CCR5- and CCR5+ kinetics. The same parameters could be used in the two groups. How sensitive are your conclusions to this issue?

We agree with the reviewers that the CCR5-/- cell kinetics could be a mixture of the CCR5- and CCR5+ kinetics. We added models that include this feature (12 new mechanistic models for T cell reconstitution part, and 4 new mechanistic ones for the section including viral dynamics) and found that the results/conclusions of the paper are not sensitive to this change:

1) We modified Equation 2 and Figure 3A to account for the possibility that CD4^+^CCR5-disrupted cells can have kinetics of CCR5- and CCR5+ cells, N_p1_ and N_p2_ compartments in the model, respectively.

a) We therefore added 12 more models including N_p2_ (models 13-24 in Figure 3—source data 2) with the same statistical assumptions for the models without N_p2_.

b) We fit the new 12 models to the T cell reconstitution data and found that the best model with compartment N_p2_ has almost the same AIC (Figure 3—source data 2 -models 11 and 23) and parameter estimates (Figure 3—source data 3) that the best model without N_p2_.

c) From this part we deduced that:

i) T cell reconstitution data could not be used to distinguish if CD4^+^CCR5-disrupted cells have a mixture of the CCR5- and CCR5+ kinetics.

ii) But regardless, the model with mixture of the CCR5- and CCR5+ kinetics didn’t differ at all from a model without that mixture.

2) When fitting viral load, we also tried to fit the model using the compartment N_p2_ (Models 5-8 in Supplementary file 5)

a) From this step we found that the model with mixture of the CCR5- and CCR5+ for disrupted cells had a significantly lower AIC (Figure 5—source data 1).

b) However model fits and comparisons between parameters and viral load steady state did not change (Figure 5, Figure 5—figure supplement 1, Figure 5—figure supplement 2, Figure 5—figure supplement 3, Figure 5—source data 2 and Figure 5—source data 3).

c) When using the best model with corresponding parameter estimates the model projections using different fp, Dose and Pr did not change either (Figure 6A-B). Minimum fp and Dose:Pr were very close to values obtained before: 76%-94%, and D:Pr>5, respectively (Figure 6C). Also, the relation of immune response (R_T_) and minimum f_p_ was the same (Figure 6D).

d) Finally, when using the model for projections of viral control for different times of ATI, we obtain the same decrease pattern of time to control.

We edited the text to include these analyses.

4) Subsection “A reduction in SHIV-specific immunity leads to higher viral rebound set points and CD4^+^CCR5+ T cell depletion following ATI in transplanted animals” and Figure 5E – this section needs more interpretation/discussion. It's surprising that the only difference in parameters between the control and treatment groups was the time to virus rebound; so presumably differences across groups derive from the steady state sizes of the different populations? Are latently infected cells depleted in the transplant groups? And it's puzzling that delayed rebound in transplant groups is taken to imply more depleted immunity. Shouldn't it imply less depletion? Also – elevated virus load after ATI only occurs in the WT transplant group, not both. Please clarify and expand this section.

Since the best model we obtained allowed distinct SHIV-specific CD8s parameter values for the ATI and pre-ART stages, we re-analyzed the data and model results using a different approach to address what drives different kinetics between the control and transplant groups.

1) We looked first at how viral load burden differed during ATI compared to pre-ART using three summary statistics: median, nadir and final viral loads from viral peak onwards for each stage (ATI vs pre-ART). Figure 4B-D now presents that result.

2) We compared each of these summary statistics per group and found that in the three summary statistics viral burden during ATI is maintained or slightly decreased for the control group, increased for the WT-transplant group (no CCR5-edition), and is increased for the CCR5-edited group but not nearly to the degree seen in the WT group.

3) Then, we compared how the model predicts that SHIV-specific immunity changed during ATI with respect to pre-ART using the parameters directly related to SHIV-specific CD8^+^ T cells (since we can only measure this with the model and not in the data). We found that SHIV-immunity change at ATI compared to pre-ART was correlated to the change in viral burden between ATI and pre-ART (median and nadir viral load during ATI:pre-ART). These results are now presented in Figure 5C-D. SHIV-immunity during ATI slightly increased for the control group (ratio over 1), decreased for the WT-transplant group, and was recovered for the CCR5-edited group

We have clarified and expanded these analyzes in the viral and T cell dynamics modeling result section.

Regarding if latently infected cells are depleted in the transplant groups, Peterson et al., 2017 and Reeves et al., 2017 support that conclusion. However, answering this question from a mathematical modeling perspective was beyond the scope of this study. For that reason, we did not explicitly have a model for the SHIV-latently infected cell compartment, or how they could have been depleted during conditioning (as we did for all other cell compartments). We only added a constant parameter ξL¯ to account for their reactivation. In the model, time to rebound was only related to latent cell reactivation (tsa=1ξL¯), but viral load kinetics after peak was related to SHIV-immunity as described above. It is possible that delayed rebound is a response to some depletion of latent cells during conditioning, and therefore, less reactivated latent cells. In that sense there is a connection of delayed rebound and depletion of SHIV-immunity during conditioning. However, exploring this relationship was beyond the scope of this study. Therefore, we have taken out the figure presenting how time to rebound was different among groups and its discussion from the manuscript to avoid confusion to the message we are presenting. Also, this result has already been presented in Peterson et al., 2018.

5) Subsection “Post-ATI viral control requires a large HSPC dose containing a high fraction of CCR5-edited cells”: Are strategies 1 and 2 not in conflict? Or does more potent conditioning somehow not necessarily imply a greater reduction in anti-SHIV immunity?

We believe they are not in conflict in the sense that we don’t mean in strategy 1 to reduce conditioning, but to find ways to augment anti-SHIV immunity despite conditioning. Several examples could include anti-SHIV CAR T cells and/or immunotherapies based on broadly neutralizing antibodies to specifically target the persistent viral reservoir but not virus-specific immune cells, and/or conditioning strategies that are targeted more specifically to HSPCs, without impacting differentiated immune subsets. In that sense we have changed the wording of strategy 1 as “augment anti-SHIV immunity despite conditioning”.

In summary, the reviewers acknowledged the analysis of an important topic by a unique dataset. However, they identified some major aspects that addressed the parameter identifiability within the model estimates as well as identifying the role of conditioning and dynamics of CCR5- cells. Therefore, we would like to ask you to analyze/comment on parameter identifiability for individual model estimates and validate the robustness of parameter estimates and, hence, model selection. This will be subject to re-review to determine if these issues have been satisfactorily addressed.

We appreciate the reviewer’s comments and acknowledgements and believe we have addressed to the best of our abilities the major aspects regarding parameter identifiability/model selection, role of conditioning and dynamics of CCR5- cells.

[Editors' note: further revisions were suggested prior to acceptance, as described below.]

The manuscript has been improved but the reviewers have identified some remaining issues that need to be addressed before acceptance, as outlined below:1) Figure 2—figure supplement 1, Panel B: The statistical comparison total vs. TN+TCM does not make sense. As I assume that Total=T_N_+T_CM_+T_EM_, "Total" does not contain additional information when there is already a comparison T_N_+T_CM_ vs. T_EM_.

We have changed Figure 2—figure supplement 1, Panel B: taking out the statistical comparison for total vs. TN+TCM

2) The manuscript contains a lot of typos and needs proofreading.

We have proofread the manuscript and took care of the typos we found.

3) The Berlin and London patients are still not described.

We have included a description of the Berlin and London patients in the Introduction.

4) The authors should provide all data and code used to perform the analyses.

Besides of the code already submitted we have provided the code for analyses in Figure 2, Figure 3, Figure 4, Figure 5 and Figure 6.